# Reshape-then-Factorize: Communication-Efficient FL via Model-Agnostic Projection Optimization

## Abstract

Federated learning (FL) enables collaborative model training across distributed clients without sharing sensitive data. However, communication overhead remains a significant bottleneck, particularly for large-scale models. Low-rank decomposition techniques address this by approximating each layer's weights or gradients with a product of low-rank matrices, thereby reducing the communication cost in FL. While effective, these methods are constrained by the layer's architecture and shapes, limiting their flexibility and performance. We propose *Model-Agnostic Projection Optimization* (MAPO), a novel method that reshapes and factorizes the full model gradient into a *fixed reconstruction matrix* and a *trainable projection vector*, avoiding layer-wise decomposition and architecture constraints. MAPO directly optimizes the projection in a randomly sampled subspace, with all clients generating the reconstruction matrix via a shared random seed, incurring no additional communication overhead for synchronization. By decoupling the gradient from architectural constraints through reshaping and enabling communication-free exploration of dynamic subspaces via seed sharing, MAPO provides a more flexible and efficient low-rank representation. Empirical results demonstrate the effectiveness of MAPO in various FL settings.

## 1 Introduction

Federated Learning (FL) is a distributed framework that enables model training across many clients without centralizing data. In each communication round, clients download a global model, update it using local data, and send modifications back to the server, which aggregates them (e.g., via FedAvg [1]). While this iterative process enables collaborative learning, frequent transmission of model updates incurs significant communication overhead, limiting FL application, particularly with large models or resource-constrained clients.

Communication-Efficient Federated Learning (CEFL) literature [2] proposes a vast range of strategies to reduce communication load. These methods are typically categorized into *sketched updates*, which compress the total model update after optimization (e.g., subsampling, quantization, random projection), and *structured updates*, which restrict the trainable parameters to a lower-dimensional subspace before optimization (e.g., random masks, weight-sharing, and low-rank decomposition) [3].

**Low-rank decomposition** is a widely used approximation technique that expresses model gradients or parameters as the product of low-rank matrices [4]. *Parameter decomposition* is particularly effective for Parameter-Efficient Fine-Tuning (PEFT), where auxiliary low-rank adaptation (LoRA) modules are added to each layer to reduce computation and storage overhead of full-model fine-tuning [5]. Although LoRA alleviates communication burdens in FL, constraining model parameters to a low-rank subspace can degrade performance. In contrast, *gradient decomposition* preserves full-rank model representations during inference and restricts only the gradients to a low-rank form during backpropagation [6–10]. A visual comparison is shown in Figure 1.

Submitted to 39th Conference on Neural Information Processing Systems (NeurIPS 2025). Do not distribute.

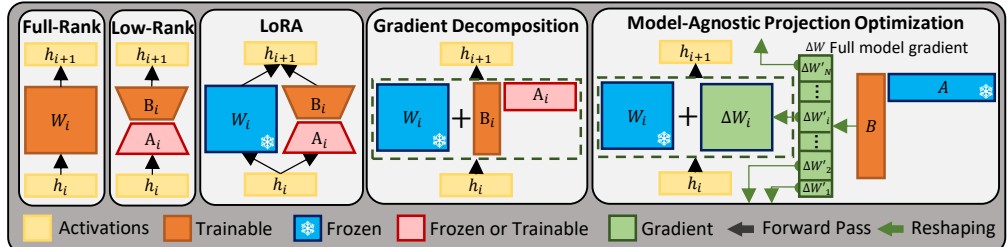

Figure 1: Comparison of various decomposition methods, from left: no decomposition, low-rank parameter decomposition, frozen model with low-rank adapter (LoRA), low-rank gradient decomposition, and MAPO.

**Challenges.** While CEFL methods for gradient decomposition [11–15], parameter decomposition [16–20], or LoRA variants [21–25] offer notable benefits, they face several key challenges: 1) The layer-wise decomposition that adheres to the structural constraints (e.g., fully connected or convolutional), requiring *architecture-dependent* implementation for each layer decomposition. 2) Given a decomposition $\Delta W_i \in \mathbb{R}^{d_1 \times d_2} \approx B_i A_i$, where $A_i \in \mathbb{R}^{r \times d_2}$ and $B_i \in \mathbb{R}^{d_1 \times r}$, the number of transmitted parameters is $\mathbf{C} = |A_i| + |B_i| = r(d_1 + d_2)$ for $r \in \mathbb{N}$, restricting the communication rate to multiples of $(d_1 + d_2)$, imposing a *rigid communication granularity* as $C \in (d_1 + d_2)\mathbb{N}$. 3) Given $M$ number of clients and $(A_i^j, B_i^j)$ denoting the low-rank decomposition of layer $i$ from client $j$, averaging these low-rank matrices is *not equivalent to full-rank aggregation* as:

$$\frac{1}{M}(B_i^1 A_i^1 + B_i^2 A_i^2 + \cdots + B_i^M A_i^M) \neq \frac{1}{M}(B_i^1 + B_i^2 + \cdots + B_i^M)\frac{1}{M}(A_i^1 + A_i^2 + \cdots + A_i^M).$$

4) Although fixing all $\{A_i^j\}_{j=1}^M$ matrices to the same values can mitigate the aggregation problem and improve the communication granularity to $\mathbf{C} \in d_1\mathbb{N}$, as shown in FA-LoRA [21] and EvoFed [26], it restricts the model's ability to explore richer subspaces, often leading to *suboptimal solutions* [25]. Thus, we aim to answer the following key question:

*How can we develop an architecture-independent model-wide decomposition that offers flexibility on communication rate, address the low-rank averaging problem, and suboptimality of freezing $A$?*

**Key Ideas.** We propose a novel Model-Agnostic Projection Optimization (**MAPO**) that streamlines gradient projection and addresses its challenges while being computationally lighter than layer-wise methods. Our key ideas are described as follows:

**(i)** Firstly, MAPO reimagines low-rank gradient projection by treating the entire model gradient as a single matrix rather than layer-by-layer decomposition. It eliminates architecture-specific constraints by merging the flattened gradients of all layers, constructing the *universal gradient vector* $\Delta W \in \mathbb{R}^d$.

**(ii)** Secondly, given any communication budget $k$, MAPO pads $\Delta W$ with zeros so the length becomes divisible by $k$. Afterwards, padded $\Delta W$ will be reshaped to $\Delta W' \in \mathbb{R}^{k \times \lceil d/k \rceil}$ which further can be decomposed it into a $A \in \mathbb{R}^{1 \times \lceil d/k \rceil}$ and $B \in \mathbb{R}^{k \times 1}$ matrices, as $\Delta W' = BA$.

**(iii)** Lastly, instead of relying on a fixed $A$, MAPO explores new subspaces in each federated round through reinitialization of $A$, mitigating the risk of suboptimal convergence. Synchronization of $A$ is achieved efficiently via a shared seed, removing the need to transmit $A$.

**Summary of Contributions.** By integrating **(i)** model-level decomposition, **(ii)** flexible communication rate, and **(iii)** subspace exploration, MAPO offers a flexible trade-off between communication cost and performance while remaining more efficient than low-rank decomposition methods. Figure 3 illustrates the distinction between MAPO and other paradigms. Our main contributions are:

- We introduce model-agnostic optimization of gradient projections that enhances communication and computation efficiency, boosts performance through exploration, and offers more flexibility in balancing communication and error rate.

- We provide theoretical analysis for MAPO convergence behavior, and establish its computation efficiency compared to layer-wise factorization with the same communication and error rates.

- We conduct extensive experiments across diverse datasets, model architectures, and baselines, demonstrating that MAPO surpasses existing methods in full training and fine-tuning scenarios.

## 2  Background and Related Works

In this section, we review key CEFL approaches in relation to MAPO. We begin with sketched update techniques that project model updates into subspaces, outlining their limitations. Then, we examine structured update methods, particularly projection optimization, highlighting the unique opportunities and challenges introduced by operating within a fixed subspace.

## 2.1 Sketched update vs. Structured update

**Sketched update** includes techniques such as sparsification [3], quantization [27–33], gradient subspace projection [34–36], and random subspace projection [26, 37]. They aim to compress the information in the update vector $\Delta W \in \mathbb{R}^d$ defined as the difference between the locally optimized and the global model $\Delta W = W^* - W_g$, where $W^*$ can be the result of multiple local epochs.

The subspace projection process [37–40] defines a random matrix $A \in \mathbb{R}^{p \times d}$, and finds the projection vector $B \in \mathbb{R}^p$, which minimizes the reconstruction error $\|\Delta W - BA\|_2$, where $d$ denotes the total number of model parameters and $p \ll d$ is compressed length:

$$B^* = \arg\min_{B \in \mathbb{R}^p} \|\Delta W - BA\|_2 \quad ; \quad B^* \approx \Delta W \mathbf{A}^\top (\mathbf{A}\mathbf{A}^\top)^{-1}.$$

As the matrix $A$ is considerably large ($p \times d$), various methods propose novel designs for $A$ to adapt it for large-scale models. Notably, defining $A$ as a subset of seen gradient vectors results in a significantly lower rank of $A$ suffices for an effective projection [34–36]. More recently, EvoFed [26] utilizes evolutionary strategies to evolve $A$, improving its representation and efficiency.

**Sketching Limitations.** Although sketched methods benefit from a full-rank training, their shortcoming is blindness to the loss surface $\mathcal{L}(W; \mathcal{D})$ and alternative solutions besides $\Delta W$ that can be reconstructed from the projection subspace. They typically perform well, given a sufficient communication budget, but as the compression rate increases, the reconstruction of the projection vector ends up far off from $\Delta W$. In contrast, subspace optimization directly finds the steepest direction within the subspace, leading to a more effective reduction in loss. Figure 2 presents an example of centralized MNIST training, illustrating the performance degradation of sketched update techniques such as EvoFed [26] and Top-$k$ Sparsification [3] compared to MAPO. As sparsity increases, MAPO continues to converge, even having 2 or 4 trainable parameters out of 11,274.

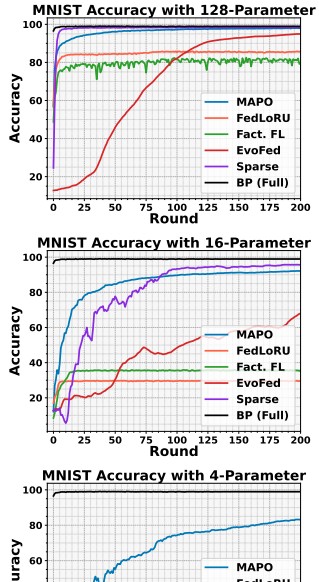

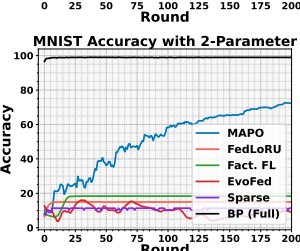

Figure 2: MNIST performance for varying trainable parameters.

**Structured update** techniques reduce the number of trainable parameters and communication cost by constraining the weights or gradients to a low-rank subspace by structural modification such as pruning [41–44], weight–sharing [45–47], low-rank gradient [11–15], and parameter decomposition [16–20], including LoRA and its variants [5, 21–24]. Although parameter decomposition techniques reduce the model size and representation, resulting in subpar performance for general training, as shown in Figure 2 for Factorized-FL [18]. Therefore, CEFL generally adopts a gradient decomposition direction. In particular, gradient decomposition methods with freezing $A$, also known as *projection optimization*, remain popular owing to strong theoretical foundations, reduced communication, and hardware friendliness [6–10].

Prior works on gradient decomposition relied on each layer's shape and architecture, producing a unique $A_i$ and $B_i$ matrices for each layer, limiting the feasibility of sharing a projection matrix $A$ across layers. MAPO overcomes this limitation by evenly partitioning the whole model gradient vector $\Delta W \in \mathbb{R}^d$ into $k$ segments $\{\Delta W_i'\}_{i=1}^k \in \mathbb{R}^{k \times \lceil d/k \rceil}$, allowing the use of a shared random reconstruction matrix $A \in \mathbb{R}^{1 \times \lceil d/k \rceil}$ across all partitions, maintaining the benefits of model-wide projection while substantially reducing memory costs.

## 2.2 Parameter-efficiency vs. Communication-efficiency

Despite their apparent similarities, parameter decomposition and gradient decomposition methods differ fundamentally in assumptions and objectives. Parameter decomposition directly imposes a low-rank structure on the model parameters, effectively replacing the original model with a compressed version. Although this reduces the total number of parameters and computational overhead, it still requires transmitting all parameters at each communication round, resulting in no relative reduction in communication per parameter. In contrast, gradient decomposition methods maintain the original

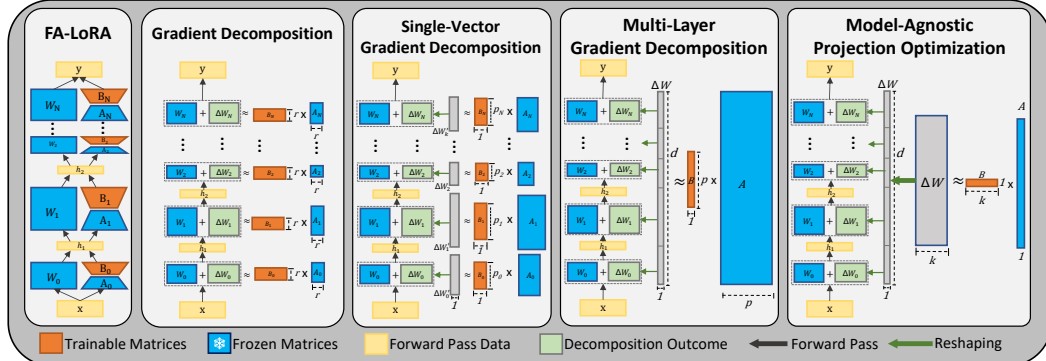

Figure 3: Step-by-Step illustration of methodology based on propositions, demonstrating how each step will contribute to designing MAPO factorization and differing from LoRA architecture.

model architecture and computational complexity but substantially reduce communication overhead by transmitting compressed updates that are significantly smaller than the full model.

In this work, to ensure a fair assessment of communication efficiency, we evaluate MAPO against gradient-based compression baselines under consistent model architectures. Additional experiments with parameter decomposition and LoRA-based methods are provided in Appendices B and C for completeness. Key methodological distinctions among related works are summarized in Table 1.

Table 1: Summary of CEFL methods and objectives. The column "Comm. Flex" indicates support for arbitrary bitrates, and "Agg. Eq." denotes equivalence between low-rank and full-rank averaging.

| Method | Scope | Target | Full-rank Inference | Agg. Eq. | PEFT | Fixed Subspace | Arch-Agnostic | Comm Flex | Personalized FL |
|---|---|---|---|---|---|---|---|---|---|
| Sparsification [3] | Model | Update | ✓ | ✓ | ✗ | ✗ | ✓ | ✓ | ✗ |
| Quantization [32] | Model | Update | ✓ | ✓ | ✗ | ✗ | ✓ | ✓ | ✗ |
| EvoFed [26] | Model | Update | ✓ | ✓ | ✗ | ✓ | ✓ | ✓ | ✗ |
| Factorized-FL [18] | Layer | Parameter | ✗ | ✗ | ✗ | ✗ | ✗ | ✗ | ✓ |
| LoRA [5] | Layer | Adapter | ✗ | ✗ | ✓ | ✗ | ✗ | ✗ | ✗ |
| FA-LoRA [21] | Layer | Adapter | ✗ | ✓ | ✓ | ✓ | ✗ | ✗ | ✗ |
| SA-LoRA [25] | Layer | Adapter | ✗ | ✗ | ✓ | ✗ | ✗ | ✗ | ✓ |
| FedLoRU [13] | Layer | Gradient | ✓ | ✓ | ✗ | ✓ | ✗ | ✗ | ✗ |
| **MAPO (Ours)** | Model | Gradient | ✓ | ✓ | ✗ | ✓ | ✓ | ✓ | ✗ |

# 3 Proposed Method

In this section, we introduce MAPO and its application in FL. We first present the MAPO factorization technique and discuss its key properties regarding communication efficiency and error rate. Subsequently, we detail how MAPO can be effectively integrated into the FL training process.

## 3.1 Model-Agnostic Projection Optimization (MAPO)

**MAPO Description.** MAPO performs a black-box, model-agnostic factorization of the global model gradient $\Delta W \in \mathbb{R}^d$, avoiding architecture-specific constraints and enabling continuous subspace exploration during optimization. Specifically, MAPO partitions $\Delta W$ into $k$ segments $\{\Delta W_i'\}_{i=1}^k \in \mathbb{R}^{k \times \lceil d/k \rceil}$ and employs a shared random reconstruction matrix $A \in \mathbb{R}^{1 \times \lceil d/k \rceil}$ across all partitions. This design preserves model-wide projection benefits while substantially reducing memory overhead. As illustrated in Figure 1, MAPO reshapes the universal gradient $\Delta W \in \mathbb{R}^{d \times 1}$ into $\Delta W' \in \mathbb{R}^{k \times \lceil d/k \rceil}$, which is then decomposed into a reconstruction vector $A$ and a projection vector $B \in \mathbb{R}^{k \times 1}$. Figure 3 shows a step-by-step visualization analogous to Theorems 3.4 to 3.6.

**MAPO Properties.** MAPO aims to construct an expressive subspace, enabling a small $B$ to encode sufficient information for updating the model efficiently. First, we formally define the concepts of communication overhead rate and reconstruction error rate in the context of matrix factorization in Theorems 3.2 and 3.3. Using these definitions, Theorem 3.4 establishes that reshaping a single layer preserves both the factorization error and communication rates. Extending this, Theorem 3.5 demonstrates that vectorizing multiple layers into a single matrix similarly maintains these properties. Finally, this leads to the proof of Theorem 3.6, which introduces a computationally and communication-efficient, model-agnostic factorization method as an alternative to traditional layer-wise gradient projection techniques. Appendix G presents the formal proofs.

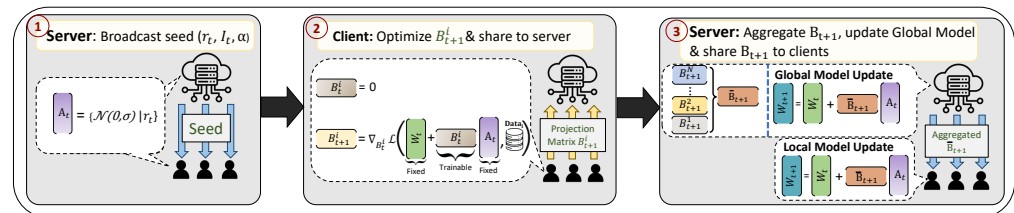

Figure 4: Application of MAPO to communication-efficient FL.

**Assumption 3.1** (**Gaussian Matrices are Full Rank**). Let $A \in I\!R^{m \times n}$ be a random matrix with entries drawn independently from a Gaussian distribution $\mathcal{N}(0, \sigma^2)$. Then, $A$ is almost surely of full rank, i.e., $\text{rank}(A) = \min(m, n)$, as the probability of $A$ being rank deficient is zero. This result follows from standard properties of random matrices [48, 49].

**Definition 3.2** (**Communication Overhead Rate**). Let $\Delta W_i \in I\!R^{d_1 \times d_2}$ be the update matrix of a model. Suppose the factorization of $\Delta W_i$ as $\Delta W_i = B_i A_i$, where $A_i \in I\!R^{q \times d_2}$ is a fixed random matrix and $B_i \in I\!R^{d_1 \times q}$ is a trainable matrix with $q \leq \min(d_1, d_2)$ being the factorization rank. The **communication overhead rate** $\text{CO}_{rate}$ is defined as the ratio of the size of $B_i$ to the size of $\Delta W$:

$$\text{CO}_{rate} = \frac{\text{size}(B_i)}{\text{size}(\Delta W_i)} = \frac{q}{d_2}.$$

**Definition 3.3** (**Reconstruction Error Rate**). Using the same factorization as Theorem 3.2, the *reconstruction error rate* is the expected ratio of the reconstruction error to the original model update. Given full-rank random reconstruction (Theorem 3.1), it is expressed as:

$$\frac{\mathbb{E}_{A_i}\left[\|\Delta W_i - B_i A_i\|_2^2\right]}{\|\Delta W_i\|_2^2} = 1 - \frac{q}{d_2}.$$

**Proposition 3.4** (**Single-Vector Factorization**). *Let $\Delta W_i$, $A_i$, and $B_i$ be factorizations of a single layer of the network as in Theorem 3.2. By reshaping $\Delta W_i$ into $\Delta W_i' \in I\!R^{1 \times d_1 d_2}$ the factorization of $\Delta W_i' = B_i' A_i'$ where $A_i' \in I\!R^{p \times d_1 d_2}$ and $B_i' \in I\!R^{1 \times p}$ can achieve the same **reconstruction error** and **communication overhead** to the conventional factorization of $\Delta W_i$ when $p = q d_1$.*

**Proposition 3.5** (**Multi-Layer Factorization**). *Let $\Delta W_i$, $A_i$, and $B_i$ be **single-vector factorization** of $i$-th layer of the $N$-layered network as in Theorem 3.4. By concatenating the reshaped weights $\Delta W_i$ into $\Delta W' \in I\!R^{1 \times d}$, where $d = \sum_{i=1}^{N} d_1^i d_2^i$. The factorization of $\Delta W' = B' A'$ where $A' \in I\!R^{p \times d}$ and $B' \in I\!R^{1 \times p}$ can achieve the same **reconstruction error** and **communication overhead** to the single-vector factorization applied to each $\Delta W_i$ when $p = Nq$.*

**Proposition 3.6** (**MAPO Factorization**). *Let $\Delta W$, $A$, $B$, and rank $p$ be a multi-layer factorization of a network as defined in Theorem 3.5. By reshaping $\Delta W \in I\!R^{1 \times d}$ into $\Delta W' \in I\!R^{k \times \lceil d/k \rceil}$, and the factorization of $\Delta W' = B' A'$ where $A' \in I\!R^{1 \times \lceil d/k \rceil}$ and $B' \in I\!R^{k \times 1}$, we can achieve the same **reconstruction error** and **communication overhead** to the multi-layer factorization of $\Delta W$ when $k = p$, while reducing the memory by a factor of $k^2$.*

### 3.2 Application to Communication-Efficient Federated Learning

This subsection explains how our method, outlined in Section 3.1, is utilized in FL. The procedure pseudo-code is provided in Algorithm 1, and visualized in Figure 4.

**Matrix Construction and Broadcasting.** To ensure consistency across the network, the server and all clients start from an identical condition at each round. We guarantee identical model parameters $W_t$ and reconstruction matrix $A_t$ by broadcasting a random seed $r_t$ and the aggregated projection vector $\overline{B}_t$ at the beginning of round $t$. The initial aggregated projection vector is set to $\overline{B}_0 = \mathbf{0}$.

**In the first round** ($t = 0$), all clients and the server initialize the model $W^0$ using the same seed. The reconstruction matrix $A^0 \in I\!R^{1 \times \lceil d/k \rceil}$ is drawn from Gaussian $A^0 \sim \mathcal{N}(0, I)$, and the client $j$'s projection vector $B^{0,j} \in I\!R^{k \times 1}$ is set to 0 for all $1 \leq j \leq M$, where $M$ is the total number of clients.

**In subsequent rounds** ($t \geq 1$), clients update their local model $W^t$ using the previous round's matrix $A^{t-1}$, the model parameters $W^{t-1}$, and the broadcasted projection vector $\overline{B}^t$ as follows:

$$W^t = W^{t-1} + \mathbf{vec}(\overline{B}^t A^{t-1})_{[0:d]}, \tag{1}$$

where $\mathbf{vec}(\cdot)$ and $(\cdot)_{[0:d]}$ denotes vectorization and truncating to the first $d$ elements. Clients then regenerate $A^t \sim \mathcal{N}(0, I)$ using the seed $r^t$ and reset $B^{t,j} \leftarrow \mathbf{0}$, ensuring $A^t$ and $W^t$ synchronization.

**Algorithm 1:** Federated Learning with MAPO

**Input** : Initial random seed $r^0$, global model $W^0$, reconstruction matrix $A^0$, projection vector $\overline{B}^0$
**Output** : Final global model $W^T$

1 Initialize all clients and server with the same seed $r^0$;
2 Initialize $W^0 \in \mathbb{R}^d$, $A^0 \in \mathbb{R}^{1 \times \lceil d/k \rceil}$, $\overline{B}^0 \leftarrow \mathbf{0} \in \mathbb{R}^{k \times 1}$;
3 **for** each communication round $t = 1, \ldots, T-1$ **do**
4     **Server:** Broadcast $\overline{B}^{t-1}$ and seed $r^{t-1}$ to all clients;
5     **for** each **Client** $j = 1, \ldots, M$ (in parallel) **do**
6        Receive $\overline{B}^{t-1}$ and $r^{t-1}$;
7        Update local model: $W^t \leftarrow W^{t-1} + \mathbf{vec}(\overline{B}^t A^{t-1})[0:d]$;
8        Re-generate $A^t = \mathcal{N}(0, \sigma^2 I_d)\big|r^{t-1}$;
9        Initialize $B^{t,j} \leftarrow \mathbf{0} \in \mathbb{R}^{k \times 1}$;
10        **for** each local epoch $e = 1, \ldots, E$ **do**
11           Compute gradient: $\nabla B^{t,j} \leftarrow \nabla_{B^{t,j}} \mathcal{L}^j(W^t + \mathbf{vec}(B^{t,j} A^{t-1})[0:d], \mathcal{D}^j)$;
12           Update projection vector: $\hat{B}^{t,j} \leftarrow B^{t,j} - \eta \nabla B^{t,j}$;
13           Set $B^{t,j} \leftarrow \hat{B}^{t,j}$;
14        **end**
15        Send $\hat{B}^{t,j}$ to the server;
16     **end**
17     **Server:**
18     Re-generate $A^t = \mathcal{N}(0, \sigma^2 I_d)\big|r^{t-1}$;
19     Aggregate: $\overline{B}^t \leftarrow \frac{1}{S} \sum_{j=1}^{M} b_j \hat{B}^{t,j}$, where $S = \sum_j b_j$;
20     Update global model: $W^{t+1} \leftarrow W^t + \mathbf{vec}(\overline{B}^t A^{t-1})[0:d]$;
21     Generate new seed $r^t$ (e.g., $r^t = \text{hash}(r^{t-1})$);
22 **end**
23 **return** $W^T$;

**Local Projection Optimization.** This step optimizes the projection $\hat{B}^{t,j}$ to minimizes the client loss $\mathcal{L}(W^t + \mathbf{vec}(B^{t,j} A^{t-1})_{[0:d]}, \mathcal{D}^j)$, where $\mathcal{D}^j$ denotes client $j$'s local dataset, and model weights are derived as $W^t + \mathbf{vec}(B^{t,j} A^t)_{[0:d]}$ given the random matrix $A^t$.

At each communication round $t \geq 1$, after initializing $A_t$ and $B^{t,j}$, clients perform local training to optimize $B^{t,j}$ using their local data $\mathcal{D}^j$. The gradient of the projection vector is computed as:

$$\nabla B^{t,j} = \nabla_{B^{t,j}} \mathcal{L}^j(W^t + \mathbf{vec}(B^{t,j} A^{t-1})_{[0:d]}) \quad \textbf{for} \quad \mathcal{L}^j(W) = \frac{1}{|\mathcal{D}^j|} \sum_{x \in \mathcal{D}^j} \ell(W, x). \tag{2}$$

where $\ell(W, x)$ is the loss function (e.g., cross-entropy loss) given model $W$ and data point $x$. Therefore, given the learning rate $\eta$, only the projection $\hat{B}^{t,j}$ is updated using gradient descent as:

$$\hat{B}^{t,j} \leftarrow B^{t,j} - \eta \nabla B^{t,j}, \tag{3}$$

After optimization, clients send their optimized projection vector $\hat{B}^{t,j}$ to the server. The low dimensionality of $\hat{B}^{t,j}$ compared to $W^t$ results in communication efficiency.

**Server-Side Aggregation and Global Model Update.** Upon receiving the projection vectors $\hat{B}^{t,j}$ and their corresponding weights $b^j = |D^j|$ (e.g., batch sizes or number of local samples) from the clients, the server aggregates them to form the global projection vector:

$$\overline{B}^t = \frac{1}{S} \sum_{j=1}^{M} b^j \hat{B}^{t,j}, \quad \text{for} \quad S = \sum_{j=1}^{M} b_j \tag{4}$$

This weighted averaging captures the collective contribution of all clients, proportional to their data sizes. The server then broadcasts the aggregated projection vector $\overline{B}^t$ to all clients. After receiving $\overline{B}^t$, the server and all clients update their local models using the reconstruction matrix $A^t$ and the aggregated projection vector $\overline{B}^t$ as:

$$W^{t+1} = W^t + \mathbf{vec}(\overline{B}^t A^{t-1})_{[0:d]}. \tag{5}$$

This update integrates the clients' optimized directions into their local models and ensures synchronization across the network. This process is repeated until the global model converges.

Table 2: Summary of datasets and models used in our experiments.

| Dataset | Client Distribution | Train/Test | # Classes | Model | # Parameters |
|---|---|---|---|---|---|
| MNIST [50] | Non-IID (2 classes) | 60K / 10K | 10 | CNN - 2 Layers | 11,274 |
| FMNIST [51] | Non-IID (2 classes) | 60K / 10K | 10 | CNN - 2 Layers | 11,274 |
| CIFAR-10 [52] | Non-IID (2 classes) | 50K / 10K | 10 | CNN - 4 Layers | 1,146,634 |
| CIFAR-100 [52] | Non-IID (10 classes) | 50K / 10K | 100 | WideResNet 16d4w | 2,854,420 |
| TinyImageNet [53] | Non-IID (10 classes) | 100K / 10K | 200 | WideResNet 16d4w | 2,880,120 |
| Shakespeare [54] | Distributed by Roles | 14K / 2K | 65 | LSTM | 814,957 |
| Sentiment140 [54] | Distributed by Users | 1.4M / 200K | 2 | Transformer | 2,221,570 |
| GLUE Tasks [55] | Non-IID | *differ per task* | *differ per task* | RoBERTa-Large | 357,199,876 |

## 4  Convergence Analysis

We analyze the convergence behavior of FL with MAPO.

**Assumption 4.1.** For each $j$, $\mathcal{L}^j(v)$ is $\beta$-smooth, i.e., $\left\|\nabla\mathcal{L}^j(u) - \nabla\mathcal{L}^j(v)\right\| \leq \beta\|u-v\|$ for any $u, v$.

**Assumption 4.2.** Variance of the stochastic gradient of $D^j$ is bounded for each client $j$, i.e.,

$$\mathbb{E}\left[\left\|\nabla\mathcal{L}^j(W) - \widetilde{\nabla}\mathcal{L}^j(W)\right\|^2\right] \leq \sigma_l^2$$

**Theorem 4.3.** *Let the learning rate satisfy* $\eta_t \leq \frac{1-4\epsilon}{4\beta(1+\epsilon)}$. *Then, the algorithm achieves the bound:*

$$\frac{1}{4H_T}\sum_{t=0}^{T-1}\eta_t\mathbb{E}\left[\left\|\nabla\mathcal{L}(W^t)\right\|^2\right] \leq \frac{\mathbb{E}\left[\mathcal{L}(W^0)\right]-\mathcal{L}^*}{H^T} + 2(\epsilon+\beta+\beta\epsilon)\sigma_l^2\frac{1}{H^T}\sum_{t=0}^{T-1}\eta_t^2,$$

*where* $H_T = \sum_{t=0}^{T-1}\eta_t$, $\epsilon$ *is JL Lemma distortion parameter, and* $\mathcal{L}^*$ *is the minimum value of* $\mathcal{L}(W)$.

With a decreasing learning rate satisfying $\sum_{t=0}^{\infty}\eta_t \to \infty$, $\sum_{t=0}^{\infty}\eta_t^2 < \infty$ ($\eta_t = \frac{\eta_0}{t+c}$ for some constants $\eta_0 > 0$, $c > 0$), the term $H_T = \sum_{t=0}^{T-1}\eta_t$ grows unbounded, while the weighted sum $\sum_{t=0}^{T-1}\eta_t^2$ remains finite. Therefore, the right-hand side of Theorem 4.3's bound satisfies:

$$\frac{\mathbb{E}[\mathcal{L}(W^0)]-\mathcal{L}^*}{H_T} \to 0, \quad \frac{1}{H_T}\sum_{t=0}^{T-1}\eta_t^2 \to 0 \quad \text{as} \quad T \to \infty.$$

Thus, confirming convergence to a stationary point, as the gradient norm average satisfies:

$$\frac{1}{H_T}\sum_{t=0}^{T-1}\eta_t\mathbb{E}\left[\|\nabla\mathcal{L}(W^t)\|^2\right] \to 0,$$

As shown above, the convergence bound is influenced by the factor $\epsilon+\beta+\beta\epsilon$. In particular, the bound becomes tightest and achieves the highest communication efficiency when there is no reconstruction error, i.e., when $\epsilon = 0$. The complete proof of Theorem 4.3 is located in Appendix H.

## 5  Experimental Setup

We evaluate MAPO across diverse model architectures, tasks, and baselines. The benchmarks span five image classification datasets—MNIST [50], FMNIST [51], CIFAR-10, CIFAR-100 [52], and TinyImageNet [53]—as well as sequential tasks, including next-character prediction on Shakespeare and sentiment analysis on Sentiment140, both drawn from the LEAF benchmark suite [54], tailored for FL. Additionally, we evaluate MAPO as a fine-tuning method, alongside LoRA baselines on various GLUE [55] tasks, highlighting the communication and computation efficiency in Appendix B. The dataset specifications and corresponding model architectures are summarized in Table 2, highlighting MAPO's adaptability across varying data modalities, model scales, and application domains.

**Non-IID Distribution.** To simulate realistic FL conditions, we partition the training datasets in a non-IID manner across 100 clients. For image classification and GLUE tasks, each client is assigned a distinct subset of classes. For LEAF tasks, we follow the natural user-based partitioning, where individual Shakespearean roles and Twitter users correspond to separate clients.

**Model Architectures.** We evaluate MAPO across diverse architectures of varying complexity, including CNNs (2-layer for MNIST and FMNIST; 4-layer for CIFAR-10), WideResNet (width 4, depth 16) for CIFAR-100 and TinyImageNet, LSTM for next-character prediction, Transformer for sentiment analysis, and RoBERTa for GLUE tasks. Detailed architecture specifications and hyperparameters are in Appendix D.

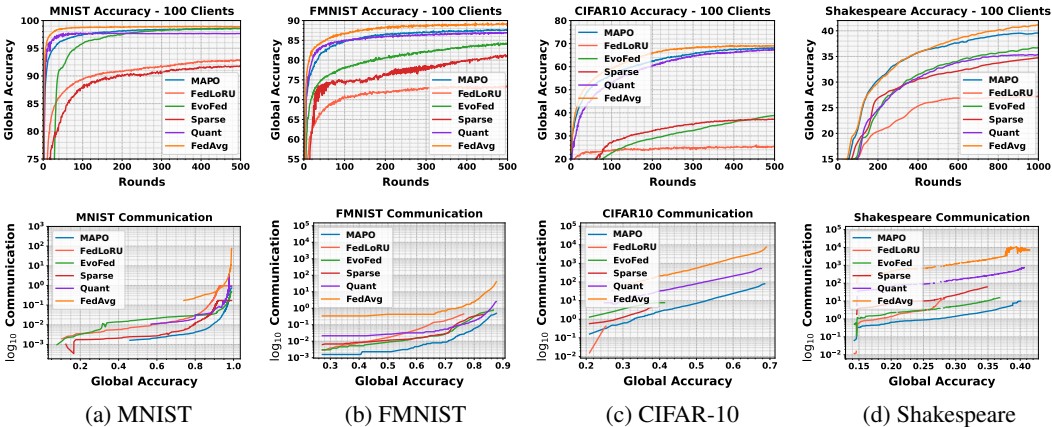

| | | | (a) MNIST | | (b) FMNIST | | (c) CIFAR-10 | | (d) Shakespeare |

Figure 5: **Performance comparison** of all methods on MNIST, FMNIST, CIFAR-10, and Shakespeare datasets. The top row shows the accuracy, while the bottom row illustrates the communication cost per accuracy.

Table 3: Summary of maximum accuracy (%) and communication cost (% relative to FedAvg). Accuracy values report mean (±std) over 3 runs, estimated from observed variance.

| | MNIST | | FMNIST | | CIFAR-10 | | CIFAR-100 | | Shakespeare | | Sent140 | | TinyImageNet | |
|---|---|---|---|---|---|---|---|---|---|---|---|---|---|---|
| Method | Com. | Acc. | Com. | Acc. | Com. | Acc. | Com. | Acc. | Com. | Acc. | Com. | Acc. | Com. | Acc. |
| FedAvg | 100 | 98.9 (±0.1) | 100 | 89.2 (±0.2) | 100 | 69.0 (±0.2) | 100 | 43.47 (±0.3) | 100 | 41.86 (±0.3) | 100 | 74.90 (±0.3) | 100 | 36.48 (±0.4) |
| Sparse | 15.3 | 92.1 (±0.4) | 24.1 | 81.1 (±0.4) | 2.7 | 37.15 (±0.5) | 1.20 | 33.72 (±0.5) | 1.73 | 34.86 (±0.4) | 1.93 | 74.21 (±0.3) | 1.32 | 25.34 (±0.5) |
| Quantize | 31.3 | 97.6 (±0.2) | 24.1 | 87.1 (±0.3) | 15.2 | 67.40 (±0.3) | 6.10 | 40.05 (±0.4) | 10.11 | 35.45 (±0.4) | 13.85 | 73.70 (±0.3) | 8.75 | 34.47 (±0.4) |
| EvoFed | 9.40 | 98.5 (±0.2) | 7.60 | 84.7 (±0.3) | 3.4 | 39.50 (±0.4) | 20.4 | 37.62 (±0.4) | 0.23 | 36.76 (±0.3) | 0.40 | 70.50 (±0.3) | 1.85 | 15.40 (±0.5) |
| FedLoRU | 30.2 | 93.8 (±0.4) | 17.9 | 74.1 (±0.5) | 1.7 | 23.52 (±0.5) | 1.20 | 19.10 (±0.5) | 1.67 | 28.07 (±0.5) | 1.30 | 66.61 (±0.4) | 1.27 | 7.31 (±0.5) |
| **MAPO** | **2.95** | **98.6** (±0.1) | **3.10** | **88.0** (±0.2) | **1.20** | **68.3** (±0.2) | **0.91** | **40.16** (±0.3) | **0.13** | **39.96** (±0.3) | **0.19** | **74.50** (±0.2) | **0.97** | **35.22** (±0.3) |

**Baselines.** We compare MAPO against multiple baselines, including standard compression methods with Top-$k$ subsampling (Sparse) [3], and quantization (Quant) [32]. Additionally, we evaluate MAPO against EvoFed [26], a state-of-the-art gradient compression, and FedLoRU [13], a representative gradient projection approach. Subsampling and quantization serve as references to establish MAPO's performance compared to conventional compression techniques. EvoFed provides a strong comparison to demonstrate the effectiveness of MAPO's subspace optimization relative to methods applying compression post-optimization. FedLoRU allows us to highlight MAPO's dynamic subspace exploration and its benefits over static layer-wise gradient projections. Results comparing MAPO with additional parameter-factorization (Factorized-FL [18]) and adapter-based fine-tuning baselines (LoRA [5], FA-LoRA [21], and SA-LoRA [25]) are included in Appendices B and C.

**Federated Learning Setting.** In each training round, $10\%$ of the clients are randomly selected to participate. Selected clients train locally in parallel and transmit their updates to the central server, which aggregates these updates and redistributes the resulting global model back to the clients. Model performance is evaluated centrally using the test dataset at the server.

## 6 Results and Discussions

We now discuss our experimental results in detail and provide insights into MAPO's performance. Figure 5 (top row) shows the accuracy of MAPO compared to multiple baseline methods across various datasets. MAPO consistently outperforms all other methods and achieves accuracy comparable to FedAvg, despite transmitting only a fraction of the parameters. This improvement results from MAPO's dynamic subspace optimization, which promotes effective exploration and efficient use of the communication budget to minimize the loss function directly. Additionally, Figure 5 (bottom row) illustrates the minimal communication cost required by each method to reach a given accuracy level, highlighting MAPO's significantly lower communication demands (logarithmic scale on the y-axis). Additional results on CIFAR-100, TinyImageNet, and Sentiment140 are presented in Appendix A.

Table 3 summarizes experimental results by comparing the maximum accuracy of each baseline and their communication cost relative to FedAvg. To ensure fair comparison, communication costs are reported as the percentage required to reach the accuracy of the worst-performing baseline. MAPO consistently achieves competitive accuracy with significantly lower communication overhead. Specifically, on MNIST and FMNIST, MAPO achieves 99.6% and 98.6% of FedAvg's accuracy, respectively, using only 3% of FedAvg's communication cost. For CIFAR-10, CIFAR-100, and TinyImageNet, MAPO attains 98.9%, 92.4%, and 96.5% of FedAvg accuracy, respectively, while

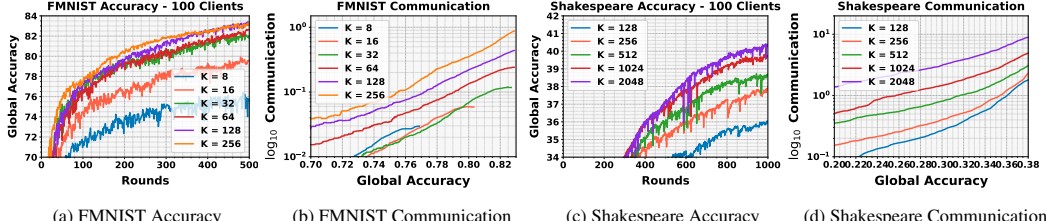

| (a) FMNIST Accuracy | (b) FMNIST Communication | (c) Shakespeare Accuracy | (d) Shakespeare Communication |

Figure 6: Accuracy and communication cost per accuracy level for FMNIST and Shakespeare datasets. Demonstrating the effect of a number of trainable parameters ($k$) on the communication efficiency of MAPO.

consuming approximately 1% of the communication. Finally, in sequential tasks (Shakespeare and Sentiment140), MAPO retains up to 95.5% and 99.5% of FedAvg's accuracy, respectively, while dramatically reducing communication to less than 0.2%.

**MAPO Hyperparameter.** MAPO simplifies gradient projection by applying a single factorization across all model parameters, thus replacing per-layer rank selection with a single hyperparameter, $k$, directly controlling communication cost and model accuracy. Figure 6 illustrates the effect of varying $k$ on performance and communication efficiency for the FMNIST and Shakespeare datasets. While a smaller $k$ significantly reduces communication overhead, it slows the convergence, requiring more training rounds. Conversely, increasing $k$ improves convergence speed and accuracy but rapidly raises communication costs, often with diminishing returns. Therefore, the optimal $k$ achieves a target accuracy with minimal total communication. Figure 6(b) and (c) show communication costs associated with specific accuracy levels, guiding the selection of optimal $k$. We use the same guidelines for all baselines to fairly tune hyperparameters.

**Fresh Reconstruction Matrix.** A key factor in MAPO's performance is using a dynamically generated reconstruction matrix $A$ rather than a fixed one. This approach promotes the exploration of new subspaces throughout training. Figure 7 illustrates the benefits of using a fresh $A$ on the FMNIST and Shakespeare datasets. We evaluate MAPO across varying numbers of trainable parameters, ranging from $2^0$ to $2^{13}$. For FMNIST, this corresponds to $0.009\%$ to $72.27\%$ of the total model parameters, while for Shakespeare, it spans from $0.0001\%$ to nearly $1\%$. In both cases, MAPO with a fresh $A$ achieves superior convergence with fewer parameters, effectively leveraging the search space. In contrast, when $A$ is frozen, performance follows a logarithmic correlation with the number of trainable parameters, requiring an exponentially larger parameter count to match the results obtained with a fresh $A$.

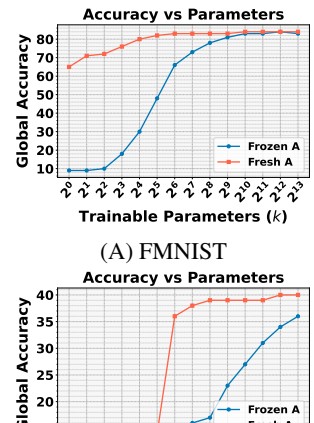

(A) FMNIST

(B) Shakespeare

Figure 7: Comparison of having a fresh $A$ vs. frozen $A$.

**Additional Results.** Comparisons with LoRA-based methods and Factorized-FL are provided in Appendices B and C. Appendix E supplements our main experiments with evaluations under IID distributions and without client sampling. Additionally, Appendix I presents a detailed memory complexity analysis, emphasizing MAPO's computational efficiency and flexibility compared to layer-wise low-rank factorization.

**Limitations.** MAPO's improved communication efficiency comes with additional computational overhead from gradient projection optimization. While significantly reduced compared to prior methods, MAPO still requires $\lceil d/r \rceil + r$ memory and computation (instead of $dr + r$; see Appendix I). MAPO complements, but does not replace, PEFT methods like LoRA, as it reduces communication overhead without decreasing the trainable parameters or storage requirements (see Appendix B).

## 7  Conclusion

We introduced *Model-Agnostic Projection Optimization* (MAPO), a novel approach for CEFL. Unlike layer-wise decomposition, MAPO factorizes the entire gradient using a projection vector and a random reconstruction matrix, regenerated at each round. MAPO balances communication efficiency and accuracy without imposing architecture-specific constraints or fixed-subspace limitations. Our theoretical analysis establishes convergence guarantees, and empirical results demonstrate superior performance and scalability across diverse datasets, confirming its practical value for FL.

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

## A  Accuracy and Communication Learning curves

This appendix provides extended experimental results that complement the main findings discussed in Section 5. We include detailed evaluations of MAPO and baseline methods on CIFAR-100, TinyImageNet, and Sentiment140 datasets. Similar to the main results, Figure 8 reports both maximum test accuracy and the communication cost required to reach a given accuracy threshold. These additional experiments further demonstrate MAPO's superior communication efficiency and consistent performance gains across more challenging and large-scale tasks.

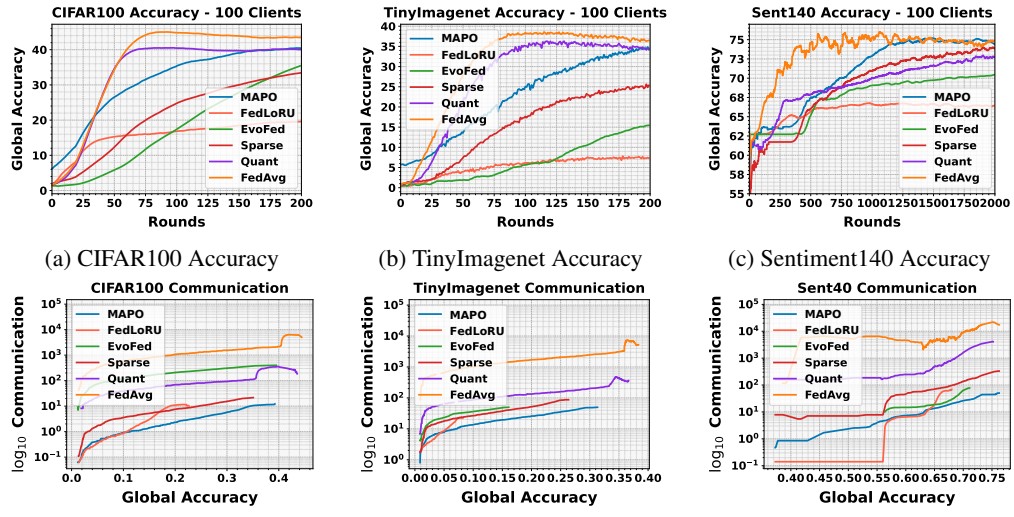

(a) CIFAR100 Accuracy     (b) TinyImagenet Accuracy     (c) Sentiment140 Accuracy

(d) CIFAR100 Comm. cost     (e) TinyImagenet Comm. cost     (f) Sentiment140 Comm. cost

Figure 8: **Performance comparison** of MAPO and baseline methods on CIFAR100, TinyImagenet, and Sentiment140 datasets. The top row shows the accuracy achieved by each method on the respective datasets, while the bottom row illustrates the communication cost associated with each method.

## B  Comparison with Low-Rank Adaptation in Fine-tuning

We conduct fine-tuning experiments using RoBERTa-large on five GLUE tasks to evaluate MAPO alongside LoRA, FA-LoRA, and SA-LoRA. Table 4 compares the number of trainable parameters and the communication load per round for each method. Table 5 summarizes fine-tuning results under federated settings, reporting communication efficiency based on the number of rounds and total communication required to reach 80% accuracy. Overall, the results indicate that MAPO improves communication efficiency without compromising performance.

Table 4: Number of trainable and communication parameters per round for different methods.

| Method | Number of trainable parameters | Number of communication parameters per round |
|---|---|---|
| LoRA | 1.83M | 0.78M |
| FA-LoRA | 1.44M | 0.39M |
| SA-LoRA | 1.83M | 0.39M |
| $\text{MAPO}_{d/1k}$ | 357M | 0.36M |
| $\text{MAPO}_{d/10k}$ | 357M | 35.70K |
| $\text{MAPO}_{d/100k}$ | 357M | 3.57K |
| $\text{MAPO}_{d/1m}$ | 357M | 357 |

Table 5: Comparison of model accuracies, communication rounds, and total communication cost.

| Model | SST2 Acc | SST2 Round | SST2 Total | QNLI Acc | QNLI Round | QNLI Total | RTE Acc | RTE Round | RTE Total | MNLIm Acc | MNLIm Round | MNLIm Total | MNLImm Acc | MNLImm Round | MNLImm Total |
|---|---|---|---|---|---|---|---|---|---|---|---|---|---|---|---|
| LoRA | 84.86 | 36 | 28.08M | 91.72 | 85 | 66.30M | 86.62 | 180 | 140.40M | 87.41 | 86 | 67.08M | 87.34 | 82 | 63.96M |
| FA-LoRA | 94.15 | 44 | 17.16M | 91.63 | 76 | 29.64M | 57.28 | — | — | 85.92 | 76 | 29.64M | 86.46 | 213 | 83.07M |
| SA-LoRA | 95.41 | 19 | 7.41M | 91.04 | 55 | 21.45M | 70.01 | — | — | 89.44 | 29 | 11.31M | 85.49 | 126 | 49.14M |
| $\text{MAPO}_{d/1k}$ | 96.79 | 5 | 1.78M | 93.14 | 11 | 3.93M | 87.91 | 23 | 8.21M | 88.90 | 17 | 6.07M | 88.26 | 22 | 7.85M |
| $\text{MAPO}_{d/10k}$ | 96.10 | 5 | 178.50K | 92.57 | 8 | 285.60K | 89.57 | 23 | 821.10K | 88.81 | 18 | 642.60K | 87.43 | 25 | 892.50K |
| $\text{MAPO}_{d/100k}$ | 95.53 | 5 | 17.85K | 89.24 | 7 | 24.99K | 84.38 | 24 | 85.68K | 85.04 | 20 | 71.40K | 84.60 | 29 | 103.53K |
| $\text{MAPO}_{d/1m}$ | 90.37 | 7 | 2.50K | 80.09 | 34 | 12.14K | 57.04 | — | — | 72.46 | — | — | 37.76 | — | — |

## C  Comparison with Factorized-FL

In this section, we present a detailed comparison between MAPO and Factorized-FL as a representative of the parameter decomposition methods. Factorized-FL can be interpreted as a variant of rank-1 LoRA, where a sparse bias matrix substitutes for LoRA's frozen fine-tuned weights, initialized to zero. Table 6 reports the communication efficiency of MAPO and Factorized-FL on CIFAR-10 and SVHN datasets, evaluated under both IID and non-IID partitions. Each column denotes the total communication in GB required to reach X% of FedAvg's final test accuracy. Results show that MAPO achieves significantly lower communication costs compared to Factorized-FL while maintaining competitive performance across both datasets and data distributions.

Table 6: Communication cost comparison across different methods on SVHN and CIFAR-10 under IID and Non-IID settings.

| Method | SVHN | | | | CIFAR-10 | | | | Com/Round |
|---|---|---|---|---|---|---|---|---|---|
| | IID@80% | IID@90% | Non-IID@80% | Non-IID@90% | IID@80% | IID@90% | Non-IID@80% | Non-IID@90% | |
| FedAvg | 183.51 | 244.68 | 285.46 | 509.75 | 305.85 | 407.80 | 326.24 | 652.48 | 20.39GB |
| Factorized-FL | 127.75 | 182.50 | 146.00 | 219.00 | 182.50 | 292.00 | 200.75 | 310.25 | 18.25GB |
| MAPO$_{2k}$ | 0.32 | 0.79 | 0.56 | – | 0.32 | – | 0.94 | – | **0.78MB** |
| MAPO$_{16k}$ | **0.08** | **0.18** | **0.12** | **0.27** | **0.08** | **0.18** | **0.23** | **0.45** | 6.25MB |
| MAPO$_{40k}$ | 3.84 | 8.64 | 5.76 | 13.12 | 3.84 | 8.64 | 10.88 | 21.12 | 0.32GB |

## D  Implementation details and Hyperparameters

All experiments were conducted on a single NVIDIA RTX 3090 with 24 GB of memory. The main experiments and baselines are implemented with JAX [56]. The GLUE tasks and LLM fine-tuning implementation use Hugging Face libraries and models implemented in FederatedScope [57] with half precision (i.e., 16-bit float). The model configuration and training used in this work are provided in Tables 7 and 8.

Table 7: Neural network configurations for different datasets.

| Dataset | Model type | # Conv | Kernel | Hidden features | # Linear | # Output | # Parameters |
|---|---|---|---|---|---|---|---|
| MNIST | CNN | 2 | 5×5 | 8, 16 | 1 | 10 | 11.3K |
| FMNIST | CNN | 2 | 5×5 | 8, 16 | 1 | 10 | 11.3K |
| CIFAR-10 | CNN | 4 | 5×5 | 64, 64, 128, 128 | 2 | 10 | 1.1M |
| CIFAR-100 | WideResNet | 16 | 3×3 | 64×4, 128×4 | 2 | 100 | 2.8M |
| TinyImageNet | WideResNet | 16 | 3×3 | 64×4, 128×4 | 2 | 200 | 2.88M |
| Shakespeare | LSTM | - | - | 256, 8 (embed) | 2 | 65 | 814K |
| Sentiment140 | Transformer | - | - | 512, 96 (embed) | 2 | 2 | 2.2M |
| SVHN | CNN | 4 | 5×5 | 64, 64, 128, 128 | 2 | 10 | 1.1M |
| GLUE | RoBERTa-large | - | - | 1024 (hidden) | 2 | Varies | 357M |

Table 8: Training hyperparameters for FedAvg and variants.

| Hyperparameter | MNIST | FMNIST | CIFAR-10 | CIFAR-100 | TinyImageNet | Sentiment140 | Shakespeare | SVHN | GLUE |
|---|---|---|---|---|---|---|---|---|---|
| Batch size | 32 | 32 | 32 | 32 | 32 | 32 | 32 | 32 | 128 |
| Optimizer | SGD | SGD | SGD | AdamW | AdamW | SGD | SGD | SGD | SGD |
| Learning rate | 0.2 | 0.2 | 0.03 | 0.1 | 0.2 | 0.001 | 0.2 | 0.03 | 0.02 |
| Momentum | 0.9 | 0.9 | 0.4 | 0.9 | 0.9 | 0.9 | 0.9 | 0.4 | 0.0 |
| L1 regularization | 0.0 | 0.0 | 1e-4 | 0.0 | 1e-5 | 0.0 | 5e-6 | 1e-4 | 0.0 |
| L2 regularization | 0.0 | 0.0 | 1e-5 | 3e-3 | 1e-4 | 0.0 | 5e-5 | 1e-5 | 0.0 |

# E   IID and Client Sampling

This section includes the results of additional experiments on IID distribution and client sampling for MNIST, FMNIST, and CIFAR-10. Across all three datasets, we observe consistent trends. Reducing the fraction of clients participating (from all clients to 10%) moderately decreases accuracy for all methods, and non-IID settings introduce additional accuracy penalties. However, MAPO's performance remains robust in these more demanding scenarios; it routinely stays close to FedAvg's high-accuracy results while maintaining significant communication savings. This resilience suggests that MAPO's approach scales well to heterogeneous data distributions and partial-participation regimes, crucial in large-scale FL deployments.

Table 9: Extrapolated MNIST results for IID vs. non-IID and full vs. 10% client participation.

| | IID | | | | Non-IID | | | |
| | All clients | | 10% clients | | All clients | | 10% clients | |
| Method | Com. | Acc. | Com. | Acc. | Com. | Acc. | Com. | Acc. |
|---|---|---|---|---|---|---|---|---|
| FedAvg | 100% | 99.6% | 100% | 99.5% | 100% | 99.3% | 100% | 98.9% |
| Sparse | 10.0% | 93.9% | 12.0% | 93.6% | 13.3% | 93.4% | 15.3% | 92.1% |
| Quantize | 22.0% | 98.8% | 25.0% | 98.5% | 29.0% | 98.2% | 31.3% | 97.6% |
| EvoFed | 6.5% | 99.4% | 7.0% | 99.2% | 8.5% | **99.0%** | 9.4% | **98.5%** |
| FedLoRU | 22.0% | 95.0% | 25.0% | 94.7% | 28.2% | 94.3% | 30.2% | 93.8% |
| MAPO | **2.0%** | **99.5%** | **2.3%** | **99.3%** | **2.7%** | **99.0%** | **2.9%** | **98.5%** |

Table 10: Extrapolated FMNIST results for IID vs. non-IID and full vs. 10% client participation.

| | IID | | | | Non-IID | | | |
| | All clients | | 10% clients | | All clients | | 10% clients | |
| Method | Com. | Acc. | Com. | Acc. | Com. | Acc. | Com. | Acc. |
|---|---|---|---|---|---|---|---|---|
| FedAvg | 100% | 91.5% | 100% | 91.0% | 100% | 90.0% | 100% | 89.2% |
| Sparse | 16.0% | 84.0% | 19.0% | 83.5% | 21.0% | 82.0% | 24.1% | 81.1% |
| Quantize | 16.0% | 89.7% | 19.0% | 89.2% | 21.0% | 88.0% | 24.1% | 87.1% |
| EvoFed | 4.5% | 87.0% | 5.5% | 86.5% | 6.8% | 85.5% | 7.6% | 84.7% |
| FedLoRU | 12.0% | 76.8% | 14.0% | 76.2% | 15.5% | 75.0% | 17.9% | 74.1% |
| MAPO | **2.0%** | **90.0%** | **2.3%** | **89.6%** | **2.7%** | **88.8%** | **3.1%** | **88.0%** |

Table 11: Extrapolated CIFAR-10 results for IID vs. non-IID and full vs. 10% client participation.

| | IID | | | | Non-IID | | | |
| | All clients | | 10% clients | | All clients | | 10% clients | |
| Method | Com. | Acc. | Com. | Acc. | Com. | Acc. | Com. | Acc. |
|---|---|---|---|---|---|---|---|---|
| FedAvg | 100% | 73.0% | 100% | 72.0% | 100% | 70.0% | 100% | 69.0% |
| Sparse | 1.8% | 41.0% | 2.0% | 40.0% | 2.4% | 38.0% | 2.7% | 37.2% |
| Quantize | 10.0% | 71.0% | 12.0% | 70.0% | 13.0% | 68.5% | 15.2% | 67.4% |
| EvoFed | 2.0% | 43.0% | 2.5% | 42.0% | 3.0% | 40.5% | 3.4% | 39.5% |
| FedLoRU | 1.1% | 27.0% | 1.3% | 26.0% | 1.5% | 24.5% | 1.7% | 23.5% |
| MAPO | **0.8%** | **71.5%** | **0.9%** | **70.8%** | **1.0%** | **69.2%** | **1.2%** | **68.3%** |

 # F   Notations

Table 12: Notation and Definitions

| Symbol | Meaning / Definition |
|---|---|
| $N$ | Number of layers in a model. |
| $i$ | Indexing notation for the layers of the model. ($1 \leq i \leq N$) |
| $M$ | Number of clients in FL. |
| $j$ | Indexing notation for clients. ($1 \leq j \leq M$) |
| $T$ | Total number of communication rounds in FL. |
| $t$ | Indexing notation for rounds. ($1 \leq t \leq T$) |
| $\mathcal{D}^j$ | Local dataset for client $j$. |
| $b^j$ | Weight for client $j$, usually set as the number of local samples $|\mathcal{D}^j|$. |
| $\Delta W$ | Model update, treated as a single vector, $\in \mathbb{R}^{d \times 1}$. |
| $W^t$ | Model parameters at communication round $t$. |
| $\overline{B}^t$ | Aggregated projection vector at round $t$, broadcast by the server. |
| $r^t$ | Random seed used to synchronize matrix generation across clients and the server. |
| $A^t$ | Reconstruction matrix at round $t$, regenerated using $r_t$. |
| $B^{t,j}$ | Trainable projection matrix for client $j$ at round $t$. |
| $\hat{B}^{t,j}$ | Locally optimized projection matrix for client $j$ at round $t$. |
| $\eta$ | Learning rate for local optimization. |
| $d$ | Total number of model parameters, defined as $d = \sum_i d_1^i d_2^i$. |
| $d_1^i, d_2^i$ | Row and column dimensions of the weight matrix for layer $i$. |
| $p$ | Factorization rank after reshaping. |
| $q$ | LoRA Factorization rank before reshaping. |
| $k$ | Design parameter controlling reshape dimension ($\Delta W'$ reshaped into $\mathbb{R}^{\lceil d/k \rceil \times k}$). |
| $A \in \mathbb{R}^{\cdot \times \cdot}, B \in \mathbb{R}^{\cdot \times \cdot}$ | Reconstruction and projection matrices in factorization. |
| $\mathcal{L}(W)$ | Global loss function. |
| $\mathcal{L}^i(W)$ | Local loss function for client $i$. |
| $\nabla \mathcal{L}(W)$ | Gradient of the global loss function. |
| $\nabla B^{t,j}$ | Gradient of local loss for the projection matrix. |
| $\sigma_l^2$ | Bounded variance of stochastic gradients. |
| $\beta$ | Smoothness constant of the loss function. |
| $\epsilon$ | Distortion parameter from the Johnson-Lindenstrauss Lemma. |

## G   Proof of Definitions and Propositions

**Definition G.1** (**Communication Overhead Rate**). Let $\Delta W \in I\!\!R^{d_1 \times d_2}$ be the update matrix of a model. Suppose the factorization of $\Delta W$ as $\Delta W = BA$, where $A \in I\!\!R^{q \times d_2}$ is a fixed random matrix and $B \in I\!\!R^{d_1 \times q}$ is a trainable matrix with $q \leq \min(d_1, d_2)$ being the factorization rank. The **communication overhead rate** $\mathrm{CO}_{rate}$ is defined as the ratio of the size of $B$ to the size of $\Delta W$:

$$\mathrm{CO}_{rate} = \frac{\mathrm{size}(B)}{\mathrm{size}(\Delta W)} = \frac{q}{d_2}.$$

**Definition G.2** (**Reconstruction Error Rate**). Using the same factorization as Theorem 3.2, the *reconstruction error rate* is the expected ratio of the reconstruction error to the original model update. Given full-rank random reconstruction (Theorem 3.1), it is expressed as:

$$\frac{\mathbb{E}_A \left[\|\Delta W - BA\|_2^2\right]}{\|\Delta W\|_2^2} = 1 - \frac{q}{d_2}.$$

*Proof.* Let $\Delta W = [\Delta w_1 \ \Delta w_2 \ \cdots \ \Delta w_{d_1}]$, where each column $\Delta w_i \in I\!\!R^{d_2}$. Similarly, the reconstruction $BA$ can be written as $[b_1 A \ b_2 A \ \cdots \ b_{d_1} A]$, where each $b_i \in I\!\!R^q$ is a trainable matrix. The reconstruction error is given by:

$$\|\Delta W - BA\|_2^2 = \sum_{i=1}^{d_1} \|\Delta w_i - b_i A\|_2^2.$$

The projection of $\Delta w_i$ onto the subspace spanned by $A$ is $P_A \Delta w_i$. The error rate $E$ is defined as:

$$E = \frac{\|\Delta w_i - \Delta w_i P_A\|_2^2}{\|\Delta w_i\|_2^2}.$$

Using the Pythagorean theorem:

$$\|\Delta w_i\|_2^2 = \|\Delta w_i P_A\|_2^2 + \|w_i - \Delta w_i \, P_A|_2^2,$$

we rewrite $E$ as:

$$E = \frac{\|\Delta w_i\|_2^2 - \|\Delta w_i P_A\|_2^2}{\|\Delta w_i\|_2^2} = 1 - \frac{\|\Delta w_i P_A\|_2^2}{\|\Delta w_i\|_2^2}.$$

The expected value of $\|\Delta w_i P_A\|_2^2$ for a full-rank random Gaussian projection is:

$$\mathbb{E}[\|\Delta w_i P_A\|_2^2] = \frac{q}{d_2}\|\Delta w_i\|_2^2.$$

Substituting this into $E$:

$$\mathbb{E}[\|\Delta w_i - b_i A\|_2^2] = 1 - \frac{\mathbb{E}[\|\Delta w_i P_A\|_2^2]}{\|\Delta w_i\|_2^2} = 1 - \frac{\frac{p}{d}\|\Delta w_i\|_2^2}{\|w_i\|_2^2} = 1 - \frac{q}{d_2}.$$

Applying this to each column $\Delta \Delta w_i$ of $\Delta W$, we obtain:

$$\mathbb{E}_A \left[\sum_{i=1}^{d_1} \|\Delta w_i - b_i A\|_2^2\right] = \sum_{i=1}^{d_1} \mathbb{E}_A \left[\|\Delta w_i - (\Delta w_i)P_A\|_2^2\right].$$

Using the expected error formula:

$$= \sum_{i=1}^{d_1} \left(1 - \frac{q}{d_2}\right)\|\Delta w_i\|_2^2 = \left(1 - \frac{q}{d_2}\right)\sum_{i=1}^{d_1}\|\Delta w_i\|_2^2.$$

Since $\|\Delta W\|_2^2 = \sum_{i=1}^{d_1}\|\Delta w_i\|_2^2$, we get:

$$\mathbb{E}_A \left[\|\Delta W - BA\|_2^2\right] = \left(1 - \frac{q}{d_2}\right)\|\Delta W\|_2^2.$$

$\square$

**Proposition G.3** (**Single-Vector Factorization**). *Let $\Delta W$, A, and B be factorizations of a single layer of the network as in Theorem 3.2. By reshaping $\Delta W$ into $\Delta W' \in I\!\!R^{1 \times d_1 d_2}$ the factorization of $\Delta W' = B'A'$ where $A' \in I\!\!R^{p \times d_1 d_2}$ and $B' \in I\!\!R^{1 \times p}$ can achieve the same **reconstruction error** and **communication overhead** to the conventional factorization of $\Delta W$ when $p = qd_1$.*

*Proof of Error Preservation.* In the single-vector setup, $\Delta W' \in I\!\!R^{d_1 d_2}$ is projected onto a subspace of dimension $p$. From random projection theory (as used in Theorem 3.3), if $A'$ is sampled such that $\text{rank}(A') = p$, then:

$$\mathbb{E}\left[\frac{\|\Delta W' - B'A'\|_2^2}{\|\Delta W'\|_2^2}\right] = 1 - \frac{p}{d_1 d_2}.$$

Substituting $p = qd_1$ gives:

$$1 - \frac{qd_1}{d_1 d_2} = 1 - \frac{q}{d_2}.$$

Hence, the expected reconstruction error satisfies:

$$\mathbb{E}\left[\|\Delta W' - B'A'\|_2^2\right] = \left(1 - \frac{q}{d_2}\right)\|\Delta W'\|_2^2,$$

which matches the original factorization. $\qquad\square$

*Proof of Communication Preservation.* For $\Delta W' \in I\!\!R^{d_1 d_2}$, with the total size $\text{size}(\Delta W') = d_1 d_2$, we have the communication overhead as:

$$\text{size}(B') = p = qd_1.$$

Thus, the communication overhead is:

$$\text{CO}'_{rate} = \frac{\text{size}(B')}{\text{size}(\Delta W')} = \frac{qd_1}{d_1 d_2} = \frac{q}{d_2},$$

which matches the original overhead.

Since both the expected reconstruction error and the communication overhead remain unchanged, the single-vector factorization with $p = qd_1$ is equivalent in terms of efficiency. $\qquad\square$

**Proposition G.4** (**Multi-Layer Factorization**). *Let $\Delta W_i$, $A_i$, and $B_i$ be **single-vector factorization** of $i$-th layer of the $n$-layered network as in Theorem 3.4. By concatenating the reshaped weights $\Delta W_i$ into $\Delta W' \in I\!\!R^{1 \times d}$, where $d = \sum_{i=1}^{n} d_1^i d_2^i$. The factorization of $\Delta W' = B'A'$ where $A' \in I\!\!R^{p \times d}$ and $B' \in I\!\!R^{1 \times p}$ can achieve the same **reconstruction error** and **communication overhead** to the single-vector factorization applied to each $\Delta W_i$ when $p = nq$.*

*Proof of Error Preservation.* For each layer $i$, a random full-rank matrix $A_i \in I\!\!R^{q \times d_2^i}$ yields an expected squared reconstruction error

$$\mathbb{E}\left[\|\Delta W_i - B_i A_i\|_F^2\right] = \left(1 - \frac{q}{d_2^i}\right)\|\Delta W_i\|_F^2.$$

Flattening $\Delta W_i$ into $\Delta W_i' \in I\!\!R^{(d_1^i d_2^i) \times 1}$, a single-vector projection of dimension $q\, d_1^i$ preserves this same error ratio (cf. Theorem 3.4).

When we concatenate all $\Delta W_i'$ into $\Delta W' \in I\!\!R^{1 \times d}$, we form a block-structured vector. Let $p := n\,q$ and let $A' \in I\!\!R^{p \times d}$ be constructed from a Gaussian distribution. By the standard random-projection argument in dimension $d$ with subspace size $p$,

$$\mathbb{E}\left[\|\Delta W' - B'A'\|_2^2\right] = \left(1 - \frac{p}{d}\right)\|\Delta W'\|_2^2.$$

Since $p = n\,q$, the overall ratio matches applying single-vector factorizations of rank $q$ to each $\Delta W_i'$ individually. $\qquad\square$

*Proof of Communication Preservation.* For each layer $i$, the single-vector factorization of $\Delta W_i$ introduces

$$\text{size}(B_i) = q\, d_1^i, \quad \text{size}(\Delta W_i) = d_1^i\, d_2^i, \quad \text{hence} \quad \frac{\text{size}(B_i)}{\text{size}(\Delta W_i)} = \frac{q}{d_1^i}.$$

574    Concatenating all $\Delta W_i'$ into $\Delta W' \in I\!R^{1 \times d}$ gives $\text{size}(\Delta W') = d$, with

$$d \;=\; \sum_{i=1}^{n} d_1^i \, d_2^i.$$

575    Meanwhile, in the multi-layer factorization, the new trainable vector $B' \in I\!R^{1 \times p}$ has

$$\text{size}(B') \;=\; p \;=\; n \, q.$$

576    Thus

$$\frac{\text{size}(B')}{\text{size}(\Delta W')} \;=\; \frac{n \, q}{\sum_{i=1}^{n} \left( d_1^i \, d_2^i \right)},$$

577    which matches the total overhead of $n$ individual rank-$q$ factorizations (one per layer) in aggregate.
578    Consequently, the communication overhead rate is also preserved.

579    Since both the expected reconstruction error (per layer or in total) and the communication overhead
580    remain the same, choosing $p = n \, q$ for $\Delta W'$ is equivalent to applying single-vector factorization of
581    rank $q$ separately to each layer. $\qquad\square$

582    **Proposition G.5** (**MAPO Factorization**). *Let $\Delta W$, A, B, and rank $p$ be a multi-layer factorization*
583    *of a network as defined in Theorem 3.5. By reshaping $\Delta W \in I\!R^{1 \times d}$ into $\Delta W' \in I\!R^{k \times \lceil d/k \rceil}$, and*
584    *the factorization of $\Delta W' = B'A'$ where $A' \in I\!R^{1 \times \lceil d/k \rceil}$ and $B' \in I\!R^{k \times 1}$, we can achieve the same*
585    ***reconstruction error** and **communication overhead** to the multi-layer factorization of $\Delta W$ when*
586    $k = p$, *while reducing the memory by a factor of $k^2$.*

587    *Proof of Error Preservation.* Since $\Delta W \in I\!R^{1 \times d}$ is reshaped into $\Delta W' \in I\!R^{k \times \lceil d/k \rceil}$, we still have
588    $\|\Delta W'\|_F^2 = \|\Delta W\|_2^2$. When $A' \in I\!R^{1 \times \lceil d/k \rceil}$ is a suitable random projection (and $B' \in I\!R^{k \times 1}$ is fit
589    accordingly), the rank-1 subspace of dimension 1 within $\lceil d/k \rceil$ induces the known expected error
590    ratio

$$\mathbb{E}\Big[\|\Delta W' - B'A'\|_F^2\Big] \;=\; \big(1 - \tfrac{1}{\lceil d/k \rceil}\big) \, \|\Delta W'\|_F^2,$$

591    since the ambient dimension is $k \times \lceil d/k \rceil \approx d$. By taking $k = p$, we obtain (via standard random-
592    projection arguments) the matching error ratio $1 - p/d$, up to negligible rounding. Therefore:

$$\mathbb{E}\Big[\|\Delta W' - B'A'\|_F^2\Big] \;=\; \big(1 - \tfrac{p}{d}\big) \, \|\Delta W'\|_F^2,$$

593                                                                   $\square$

594    *Proof of Communication Preservation.* The matrix $B' \in I\!R^{k \times 1}$ has size $k$ in total. Meanwhile,
595    $\Delta W' \in I\!R^{k \times \lceil d/k \rceil}$ has size $k \times \lceil d/k \rceil \approx d$. Thus

$$\frac{\text{size}(B')}{\text{size}(\Delta W')} \;=\; \frac{k}{\lceil d/k \rceil \, k} \;\approx\; \frac{k}{d} = \frac{p}{d}.$$

596    Setting $k = p$ matches the original ratio $\frac{p}{d}$ from $B \in I\!R^{p \times 1}$ in the multi-layer factorization. $\qquad\square$

597    *Proof of Memory Reduction by Factor $k^2$.* In standard rank-$p$ factorizations for $\Delta W \in I\!R^{1 \times d}$, one
598    typically stores a $p \times d$ projection plus a $1 \times p$ vector, whose total size scales as $dp + p$. By contrast,
599    $A' \in I\!R^{1 \times \lceil d/k \rceil}$ plus $B' \in I\!R^{k \times 1}$ has combined size $\lceil d/k \rceil + k$. When $k = p$, the ratio of these sizes
600    can be shown to drop by a factor of approximately $k^2$. Hence the approach allocates $k^2$ times less
601    memory than a naive $p \times d$ plus $1 \times p$ arrangement. As $p = k$

$$\frac{dp + p}{\lceil d/k \rceil + k} \;=\; \frac{dk + k}{\lceil d/k \rceil + k} \;\approx\; \frac{d + 1}{d/k^2 + 1} \;\approx\; k^2$$

602    Thus, the factorization $\Delta W' = B'A'$ with $k = p$ exactly preserves the original rank-$p$ error and
603    overhead while using $k^2$-fold less memory. $\qquad\square$

# H   Proof of Theorem

## H.1   Assumptions and Preliminaries

We restate the key assumptions required for the convergence analysis.

**Assumption H.1.** For each $j$, $\mathcal{L}^j(v)$ is $\beta$-smooth, i.e., $\left\|\nabla\mathcal{L}^j(u)-\nabla\mathcal{L}^j(v)\right\| \leq \beta\|u-v\|$ for any $u$, $v$.

**Assumption H.2.** Variance of the stochastic gradient of $D^j$ is bounded for each client $j$, i.e.,

$$\mathbb{E}\left[\left\|\nabla\mathcal{L}^j(W) - \widetilde{\nabla}\mathcal{L}^j(W)\right\|^2\right] \leq \sigma_l^2$$

**Lemma H.3** (Johnson-Lindenstrauss Lemma). *Given $0 < \epsilon < 1$, a set of points $\{x_1, x_2, \ldots, x_M\} \subset \mathbb{R}^d$, and a target dimension $k = O\left(\frac{\log M}{\epsilon^2}\right)$, there exists a random linear mapping $P \in \mathbb{R}^{d\times k}$ such that for all $i, j$:*

$$(1 - \epsilon)\|x_i - x_j\|^2 \leq \|x_iP - x_jP\|^2 \leq (1 + \epsilon)\|x_i - x_j\|^2.$$

In our context, the random projection matrices $B^{t,j}$ and reconstruction matrices $A^t$ satisfy the JL property with high probability.

## H.2   Proof of Theorem 1

**Theorem H.1.** *Let the learning rate satisfy $\eta_t \leq \frac{1-4\epsilon}{4\beta(1+\epsilon)}$. Then, the algorithm achieves the bound:*

$$\frac{1}{4H_T}\sum_{t=0}^{T-1}\eta_t\mathbb{E}\left[\left\|\nabla\mathcal{L}(W^t)\right\|^2\right] \leq \frac{\mathbb{E}\left[\mathcal{L}(W^0)\right]-\mathcal{L}^*}{H^T} + 2(\epsilon + \beta + \beta\epsilon)\sigma_l^2\frac{1}{H^T}\sum_{t=0}^{T-1}\eta_t^2,$$

*where $H_T = \sum_{t=0}^{T-1}\eta_t$, $\epsilon$ is JL Lemma distortion parameter, and $\mathcal{L}^*$ is the minimum value of $\mathcal{L}(W)$.*

*Proof.* By the $\beta$-smoothness of $\mathcal{L}(W)$ and taking expectation on both sides, we have

$$\mathbb{E}\left[\mathcal{L}(W^{t+1}) - \mathcal{L}(W^t)\right] \leq \mathbb{E}\left[\langle\nabla\mathcal{L}(W^t), W^{t+1} - W^t\rangle\right] + \frac{\beta}{2}\mathbb{E}\left[\left\|W^{t+1} - W^t\right\|^2\right]. \quad (6)$$

Using the update rule $W^{t+1} = W^t - \eta_t\overline{B}_tA^t$, where $\overline{B}_t = \frac{1}{M}\sum_{j=1}^M B^{t,j}$, we can rewrite the first term as:

$$\mathbb{E}\left[\langle\nabla\mathcal{L}(W^t), W^{t+1} - W^t\rangle\right] = -\eta_t\mathbb{E}\left[\left\langle\nabla\mathcal{L}(W^t), \overline{B}^tA^t\right\rangle\right]$$

$$= -\eta_t\mathbb{E}\left[\left\langle\nabla\mathcal{L}(W^t), \left(\frac{1}{M}\sum_{j=1}^M B^{t,j}\right)A^t\right\rangle\right]$$

$$= -\eta_t\mathbb{E}\left[\left\langle\nabla\mathcal{L}(W^t), \frac{1}{M}\sum_{j=1}^M B^{t,j}A^t\right\rangle\right].$$

We decompose $B^{t,j}A^t$ as:
$$\widetilde{\nabla}\mathcal{L}^j(W^t) = B^{t,j}A^t + e^{t,j},$$

where $e^{t,j} = \widetilde{\nabla}\mathcal{L}^j(W^t) - B^{t,j}A^t$ is the projection error.

Substituting back, we have:

$$\mathbf{E} = \mathbb{E}\left[\langle\nabla\mathcal{L}(W^t), W^{t+1} - W^t\rangle\right] = -\eta_t\mathbb{E}\left[\left\langle\nabla\mathcal{L}(W^t), \frac{1}{M}\sum_{j=1}^M\left(\widetilde{\nabla}\mathcal{L}^j(W^t) - e^{t,j}\right)\right\rangle\right]$$

624    Separating it into $A_1$ and $A_2$:

$$\mathbf{E} = \underbrace{-\eta_t \mathbb{E}\left[\left\langle \nabla \mathcal{L}(W^t), \frac{1}{M}\sum_{j=1}^{M}\widetilde{\nabla}\mathcal{L}^j(W^t)\right\rangle\right]}_{A_1} + \underbrace{\eta_t \mathbb{E}\left[\left\langle \nabla \mathcal{L}(W^t), \frac{1}{M}\sum_{j=1}^{M}e^{t,j}\right\rangle\right]}_{A_2}.$$

625    We will now concentrate on $A_1$ as:

$$A_1 = -\eta_t \mathbb{E}\left[\left\langle \nabla \mathcal{L}(W^t), \frac{1}{M}\sum_{j=1}^{M}\nabla \mathcal{L}^j(W^t)\right\rangle\right]$$

$$= -\frac{\eta_t}{M}\sum_{j=1}^{M}\mathbb{E}\left[\langle \nabla \mathcal{L}(W^t), \nabla \mathcal{L}^j(W^t)\rangle\right]$$

$$\underset{(a)}{=} -\frac{\eta_t}{2M}\sum_{j=1}^{M}\left\{\mathbb{E}\left[\|\nabla \mathcal{L}(W^t)\|^2\right] + \mathbb{E}\left[\left\|\nabla \mathcal{L}^j(W^t)\right\|^2\right]\right\}$$

$$+ \frac{\eta_t}{2}\mathbb{E}\left[\underbrace{\left\|\nabla \mathcal{L}(W^t) - \frac{1}{M}\sum_{j=1}^{M}\nabla \mathcal{L}^j(W^t)\right\|^2}_{=0}\right]$$

$$= -\frac{\eta_t}{2}\mathbb{E}\left[\|\nabla \mathcal{L}(W^t)\|^2\right] - \frac{\eta_t}{2M}\sum_{j=1}^{M}\mathbb{E}\left[\left\|\nabla \mathcal{L}^j(W^t)\right\|^2\right]$$

626    where (a) uses $\langle a, b\rangle = \frac{1}{2}\{\|a\|^2 + \|b\|^2 - \|a-b\|^2\}$. We now turn our attention to $A_2$ as:

627    Next, we focus on $A_2$:

$$A_2 = \eta_t \mathbb{E}\left[\left\langle \nabla \mathcal{L}(W^t), \frac{1}{M}\sum_{j=1}^{M}e^{t,j}\right\rangle\right]$$

$$\underset{(a)}{\le} \frac{\eta_t}{4}\mathbb{E}\left[\|\nabla \mathcal{L}(W^t)\|^2\right] + \eta_t \mathbb{E}\left[\left\|\frac{1}{M}\sum_{j=1}^{M}e^{t,j}\right\|^2\right]$$

$$\underset{(b)}{\le} \frac{\eta_t}{4}\mathbb{E}\left[\|\nabla \mathcal{L}(W^t)\|^2\right] + \frac{\eta_t}{M}\mathbb{E}\left[\left\|\sum_{j=1}^{M}e^{t,j}\right\|^2\right]$$

$$\underset{(c)}{\le} \frac{\eta_t}{4}\mathbb{E}\left[\|\nabla \mathcal{L}(W^t)\|^2\right] + \frac{\epsilon\eta_t}{M}\mathbb{E}\left[\left\|\sum_{j=1}^{M}\widetilde{\nabla}\mathcal{L}^j(W^t)\right\|^2\right]$$

$$\underset{(d)}{\le} \frac{\eta_t}{4}\mathbb{E}\left[\|\nabla \mathcal{L}(W^t)\|^2\right] + \frac{2\epsilon\eta_t}{M}\sum_{j=1}^{M}\left\{\mathbb{E}\left[\|\nabla \mathcal{L}^j(W^t)\|^2\right] + \mathbb{E}\left[\left\|\widetilde{\nabla}L_i(W^t) - \nabla \mathcal{L}^j(W^t)\right\|^2\right]\right\}$$

$$\underset{(e)}{\le} \frac{\eta_t}{4}\mathbb{E}\left[\|\nabla \mathcal{L}(W^t)\|^2\right] + \frac{2\epsilon\eta_t}{M}\sum_{j=1}^{M}\mathbb{E}\left[\|\nabla \mathcal{L}^j(W^t)\|^2\right] + 2\epsilon\eta_t^2\sigma_l^2$$

where (a) uses $\langle a, b \rangle \leq \frac{1}{4}\|a\|^2 + \|b\|^2$, and (b) follows Jensen's inequality, (c) comes from JL Lemma, (d) follows the inequality $\|a + b\|^2 \leq 2\|a\|^2 + 2\|b\|^2$, and (e) is based on Assumption 2. On the other hand, we can also place a bound on the second term $\mathbb{E}\left[\|W^{t+1} - W^t\|^2\right]$ as shown below:

$$
\mathbb{E}\left[\|W^{t+1} - W^t\|^2\right] = \mathbb{E}\left[\|\eta_t \overline{B_t} A^t\|^2\right] = \mathbb{E}\left[\left\|\eta_t \left(\frac{1}{M}\sum_{j=1}^{M} B^{t,j}\right) A^t\right\|^2\right]
$$

$$
\underset{(a)}{\leq} 2\eta_t^2 \mathbb{E}\left[\left\|\frac{1}{M}\sum_{j=1}^{M}\widetilde{\nabla}\mathcal{L}^j(W^t)\right\|^2\right] + 2\eta_t^2 \mathbb{E}\left[\left\|\frac{1}{M}\sum_{j=1}^{M}\left\{B^{t,j}A^t - \widetilde{\nabla}\mathcal{L}^j(W^t)\right\}\right\|^2\right]
$$

$$
\underset{(b)}{\leq} \frac{2\eta_t^2}{M}\mathbb{E}\left[\left\|\sum_{j=1}^{M}\widetilde{\nabla}\mathcal{L}^j(W^t)\right\|^2\right] + \frac{2\eta_t^2}{M}\mathbb{E}\left[\left\|\sum_{j=1}^{M}\left\{B^{t,j}A^t - \widetilde{\nabla}\mathcal{L}^j(W^t)\right\}\right\|^2\right]
$$

$$
= \frac{2\eta_t^2}{M}\mathbb{E}\left[\left\|\sum_{j=1}^{M}\widetilde{\nabla}\mathcal{L}^j(W^t)\right\|^2\right] + \frac{2\eta_t^2}{M}\mathbb{E}\left[\left\|\sum_{j=1}^{M} e^{t,j}\right\|^2\right]
$$

$$
\underset{(c)}{\leq} \frac{4\eta_t^2}{M}\sum_{j=1}^{M}\left\{\mathbb{E}\left[\|\nabla\mathcal{L}^j(W^t)\|^2\right] + \mathbb{E}\left[\left\|\widetilde{\nabla}L_i(W^t) - \nabla\mathcal{L}^j(W^t)\right\|^2\right]\right\} + \frac{2\eta_t^2}{M}\mathbb{E}\left[\left\|\sum_{j=1}^{M} e^{t,j}\right\|^2\right]
$$

$$
\underset{(d)}{\leq} \frac{4\eta_t^2}{M}\sum_{j=1}^{M}\mathbb{E}\left[\|\nabla\mathcal{L}^j(W^t)\|^2\right] + \frac{2\eta_t^2}{M}\mathbb{E}\left[\left\|\sum_{j=1}^{M} e^{t,j}\right\|^2\right] + 4\eta_t^2\sigma_l^2
$$

$$
\underset{(e)}{\leq} \frac{4\eta_t^2}{M}\sum_{j=1}^{M}\mathbb{E}\left[\|\nabla\mathcal{L}^j(W^t)\|^2\right] + \frac{2\epsilon\eta_t^2}{M}\mathbb{E}\left[\left\|\sum_{j=1}^{M}\widetilde{\nabla}\mathcal{L}^j(W^t)\right\|^2\right] + 4\eta_t^2\sigma_l^2
$$

$$
\underset{(f)}{\leq} \frac{4\eta_t^2}{M}\sum_{j=1}^{M}\mathbb{E}\left[\|\nabla\mathcal{L}^j(W^t)\|^2\right]
$$

$$
+ \frac{4\epsilon\eta_t^2}{M}\sum_{j=1}^{M}\left\{\mathbb{E}\left[\|\nabla\mathcal{L}^j(W^t)\|^2\right] + \mathbb{E}\left[\left\|\widetilde{\nabla}\mathcal{L}^j(W^t) - \nabla\mathcal{L}^j(W^t)\right\|^2\right]\right\} + 4\eta_t^2\sigma_l^2
$$

$$
\underset{(g)}{\leq} \frac{4\eta_t^2}{M}\sum_{j=1}^{M}\mathbb{E}\left[\|\nabla\mathcal{L}^j(W^t)\|^2\right] + \frac{4\epsilon\eta_t^2}{M}\sum_{j=1}^{M}\mathbb{E}\left[\|\nabla\mathcal{L}^j(W^t)\|^2\right] + 4\epsilon\eta_t^2\sigma_l^2 + 4\eta_t^2\sigma_l^2
$$

$$
= \frac{4(1+\epsilon)\eta_t^2}{M}\sum_{j=1}^{M}\mathbb{E}\left[\|\nabla\mathcal{L}^j(W^t)\|^2\right] + 4(1+\epsilon)\eta_t^2\sigma_l^2
$$

where (a), (c), and (f) are based on the inequality $\|a+b\|^2 \leq 2\|a\|^2 + 2\|b\|^2$, (b) comes from Jensen's inequality, (d), (g) derive from Assumption 2, and (e) comes from JL Lemma.

By utilizing the established bounds for $\mathbb{E}\left[\langle\nabla\mathcal{L}(W^t), W^{t+1} - W^t\rangle\right]$ and $\mathbb{E}\left[\|W^{t+1} - W^t\|^2\right]$ to Equation (6), we derive the following:

$$\mathbb{E}\left[\mathcal{L}(W^{t+1}) - \mathcal{L}(W^t)\right] \leq \mathbb{E}\left[\langle\nabla\mathcal{L}(W^t), W^{t+1} - W^t\rangle\right] + \frac{\beta}{2}\mathbb{E}\left[\|W^{t+1} - W^t\|^2\right]$$

$$\leq \underbrace{-\frac{\eta_t}{2}\mathbb{E}\left[\|\nabla\mathcal{L}(W^t)\|^2\right] - \frac{\eta_t}{2M}\sum_{j=1}^{M}\mathbb{E}\left[\left\|\nabla\mathcal{L}^j(W^t)\right\|^2\right]}_{A_1}$$

$$+ \underbrace{\frac{\eta_t}{4}\mathbb{E}\left[\left\|\nabla\mathcal{L}(W^t)\right\|^2\right] + \frac{2\epsilon\eta_t}{M}\sum_{j=1}^{M}\mathbb{E}\left[\|\nabla\mathcal{L}^j(W^t)\|^2\right] + 2\epsilon\eta_t^2\sigma_l^2}_{A_2}$$

$$+ \frac{2\beta(1+\epsilon)\eta_t^2}{M}\sum_{j=1}^{M}\mathbb{E}\left[\|\nabla\mathcal{L}^j(W^t)\|^2\right] + 2\beta(1+\epsilon)\eta_t^2\sigma_l^2$$

$$= -\frac{\eta_t}{4}\mathbb{E}\left[\|\nabla\mathcal{L}(W^t)\|^2\right]$$

$$+ \frac{\eta_t}{M}\underbrace{\left\{-\frac{1}{2} + 2\epsilon + 2\beta(1+\epsilon)\eta_t\right\}}_{\leq 0 \text{ if we choose } \eta_t \leq \frac{1-4\epsilon}{4\beta(1+\epsilon)}}\sum_{j=1}^{M}\mathbb{E}\left[\left\|\nabla\mathcal{L}^j(W^t)\right\|^2\right] + 2\eta_t^2(\epsilon + \beta + \beta\epsilon)\sigma_l^2$$

$$\leq -\frac{\eta_t}{4}\mathbb{E}\left[\|\nabla\mathcal{L}(W^t)\|^2\right] + 2\eta_t^2(\epsilon + \beta + \beta\epsilon)\sigma_l^2$$

Ultimately, by applying the telescoping sum over $t = 0, 1, \ldots, T-1$, we arrive at the following result:

$$\mathcal{L}^* - \mathbb{E}\left[\mathcal{L}(W^0)\right] \leq \sum_{t=0}^{T-1} -\frac{\eta_t}{4}\mathbb{E}\left[\|\nabla\mathcal{L}(W^t)\|^2\right] + \sum_{t=0}^{T-1} 2\eta_t^2(\epsilon + \beta + \beta\epsilon)\sigma_l^2$$

In this case, $\mathcal{L}^*$ stands for the minimum of $\mathcal{L}(W)$.

By performing a division by $H_T = \sum_{t=0}^{T-1}\eta_t$ on both sides and utilizing some algebraic adjustments, we arrive at the following expression:

$$\frac{1}{4H_T}\sum_{t=0}^{T-1}\eta_t\mathbb{E}\left[\|\nabla\mathcal{L}(W^t)\|^2\right] \leq \frac{\mathbb{E}\left[\mathcal{L}(W^0)\right] - \mathcal{L}^*}{H_T} + 2(\epsilon + \beta + \beta\epsilon)\sigma_l^2\left(\frac{1}{H_T}\sum_{t=0}^{T-1}\eta_t^2\right) \quad (7)$$

With a decreasing learning rate such as $\eta_t = \frac{\eta_0}{t+1}$, we observe that $H_T = \sum_{t=0}^{T-1}\eta_t$ tends towards infinity as $T$ grows, while $\sum_{t=0}^{T-1}\eta_t^2$ remains bounded. Therefore, as $T \to \infty$, the upper bound in Equation (7) converges to $0$, confirming the convergence to a stationary point. $\square$

# I Complexity Analysis and MAPO Flexibility

Theorems 3.4 to 3.6 discussed how the error rate and accuracy of low-rank factorization are only determined by the size of the projection vector regardless of reshaping and vectorization of layers. Although they prove that MAPO can achieve the same performance as layer-wise factorization given the same projection (communication) budget, we did not discuss the memory and computation complexity. In this section, we show that MAPO can effectively reduce computation. Furthermore, we show how layer-wise low-rank adaptation (LoRA and FA-LoRA) limits the model trade-offs and how MAPO can offer more flexibility.

## I.1 Computational Complexity

We compute the memory and computation cost for matrix allocation and multiplication in terms of standard matrix multiplication. Given matrices $A \in \mathbb{R}^{n \times m}$ and $B \in \mathbb{R}^{p \times n}$, the complexities for computing $C = BA$ are:

$$\text{Memory}_{C=AB} = O(nm + pn + pm),$$

$$\text{Time}_{C=BA} = O(mnp).$$

We aim to demonstrate that factorization under MAPO, where $W \in \mathbb{R}^{k \times \lceil \frac{d}{k} \rceil}$ is factorized into $A \in \mathbb{R}^{1 \times \lceil \frac{d}{k} \rceil}$ and $B \in \mathbb{R}^{k \times 1}$, reduces the memory and time complexity of the LoRA factorization for an $n$-layered model. In LoRA, each layer $i$ is factorized as $w_i \in \mathbb{R}^{d_i^1 \times d_i^2}$ into $A \in \mathbb{R}^{q \times d_i^1}$ and $B \in \mathbb{R}^{d_i^2 \times q}$.

We demonstrate that, given the same communication budget and factorization error rate, MAPO significantly reduces the computational cost compared to LoRA. This reduction becomes more pronounced as the number of layers or the selected rank increases. Specifically, MAPO achieves a **memory reduction** by a factor of $q^2$ and a **computation reduction** by a factor of $q$, where $q$ is the chosen LoRA rank. Furthermore, even when $q = 1$, MAPO still achieves memory savings as $\sum_{i \neq j}^{n} d_i^1 d_i^2$ scales with the number of layers. The only scenario where MAPO and LoRA yield identical efficiency is when the model consists of a single layer ($n = 1$) and a rank-1 factorization ($q = 1$).

**Memory Complexity**

Given these definitions, the memory complexities for MAPO and LoRA are:

$$\text{Memory}_{MAPO} = O\left(\left\lceil \frac{d}{k} \right\rceil + k + \left\lceil \frac{d}{k} \right\rceil k\right) \approx O\left(\frac{d}{k} + k + d\right),$$

$$\text{Memory}_{LoRA} = O\left(\sum_{i=1}^{n}(d_i^1 q + d_i^2 q + d_i^1 d_i^2)\right) = O\left(\sum_{i=1}^{n} d_i^1 q + \sum_{i=1}^{n} d_i^2 q + \sum_{i=1}^{n} d_i^1 d_i^2\right).$$

Given the same communication budget $k = \sum_{i=1}^{n} q d_i^1$ and $d = \sum_{i=1}^{n} d_i^1 d_i^2$, we rewrite LoRA's memory complexity as:

$$\text{Memory}_{LoRA} = O\left(q \sum_{i=1}^{n} d_i^2 + k + d\right).$$

For MAPO to have lower memory usage than LoRA, the following condition must hold:

$$\text{Memory}_{MAPO} \leq \text{Memory}_{LoRA},$$

$$\frac{d}{k} + k + d \leq q \sum_{i=1}^{n} d_i^2 + k + d,$$

$$\frac{d}{k} \leq q \sum_{i=1}^{n} d_i^2.$$

673 Replacing $k$ and $d$ with their respective summation terms:

$$\sum_{i=1}^{n} d_i^1 d_i^2 \leq q^2 \sum_{i=1}^{n} d_i^1 \sum_{i=1}^{n} d_i^2,$$

$$\leq q^2 \sum_{i=1}^{n} d_i^1 d_i^2 + q^2 \sum_{i \neq j}^{n} d_i^1 d_i^2.$$

674 Thus, the inequality always holds under the conditions $d_i^1, d_i^2, q, n \geq 1$, and equality occurs if
675 $q = n = 1$, which corresponds to a model with a single layer and rank-1 factorization. In this case,
676 MAPO and LoRA perform the same decomposition.

677 **Time Complexity**

678 Given the definitions, we can express the time complexities for MAPO and LoRA as follows:

$$\text{Time}_{MAPO} = O\left(\left\lceil \frac{d}{k} \right\rceil k \right) \approx O(d),$$

$$\text{Time}_{LoRA} = O\left(\sum_{i=1}^{n} q d_i^1 d_i^2 \right).$$

679 Since $d = \sum_{i=1}^{n} d_i^1 d_i^2$, we can rewrite LoRA's time complexity as:

$$\text{Time}_{LoRA} = O(qd).$$

680 For MAPO to have a lower time complexity than LoRA, the following condition must hold:

$$\text{Time}_{MAPO} \leq \text{Time}_{LoRA},$$
$$d \leq qd.$$

681 This condition is always true for $d, q \geq 1$, and equality occurs when $q = 1$.

 **I.2 MAPO Flexibility**

 Suppose our neural network has $n$ layers. Let:

$$W_i \in \mathbb{R}^{d_i^1 \times d_i^2} \quad \text{for each layer } i = 1, \ldots, n.$$

 Let $d = \sum_{i=1}^{n} d_i^1 d_i^2$ be the total number of parameters (i.e., the sum of the entries across all layers).
 Let

$$d_1 = \sum_{i=1}^{n} d_i^1.$$

 In many treatments of LoRA, the main communication or factor-size bottleneck arises from a factor
 that scales linearly with $q \cdot d_i^1$.

 **LoRA Factorization Per Layer.** LoRA factorizes each layer $W_i$ of dimension $d_i^1 \times d_i^2$ with a fixed
 rank $q$. Concretely,

$$W_i \approx W_i + B_i A_i, \qquad A_i \in \mathbb{R}^{q \times d_i^2}, \quad B_i \in \mathbb{R}^{d_i^1 \times q}.$$

 The number of additional parameters introduced by each low-rank pair $(A_i, B_i)$ is

$$\underbrace{d_i^1 \cdot q}_{\text{size of } B_i} + \underbrace{q \cdot d_i^2}_{\text{size of } A_i} = q \left( d_i^1 + d_i^2 \right).$$

 Summing over all $n$ layers,

$$\sum_{i=1}^{n} \left( d_i^1 \cdot q + q \cdot d_i^2 \right) = q \sum_{i=1}^{n} \left( d_i^1 + d_i^2 \right).$$

 Therefore, we can write the communication cost as:

$$\text{Communication cost} \approx q \sum_{i=1}^{n} d_i^1 = q \, d_1.$$

 Since $q$ must be an integer, we see that the communication overhead comes in integer multiples $d_1$, as:

$$\text{LoRA total communication} \in \{ q \, d_1 \mid q = 1, 2, \ldots \}.$$

 **There is no way to select** a non-integer $q$. Hence communication budgets strictly between $d_1$ and
 $2 \, d_1$ (or between $q \, d_1$ and $(q+1)d_1$) are not possible in layer-wise LoRA. Therefore, Any attempt to
 finely tune the communication or factor budget (e.g., to $1.5 \, d_1$) is disallowed by LoRA's integral-rank
 requirement. This **rigidity** is precisely what we seek to overcome in MAPO.

 **MAPO Factorization.** MAPO flattens or reshapes all parameters into one large matrix and then
 performs a single low-rank factorization with rank 1. A simplified abstraction is:

 1. Reshape $w_1, \ldots, w_n$ into a single matrix $W \in \mathbb{R}^{k \times \lceil d/k \rceil}$, where $d = \sum_{i=1}^{n} d_i^1 d_i^2$ is the total
 parameter count. 2. Factor $W \approx A \, B$, with

$$A \in \mathbb{R}^{1 \times \lceil d/k \rceil}, \quad B \in \mathbb{R}^{k \times 1},$$

 Once all parameters are merged, MAPO can proportionally allocate any communication budget as $k$
 can be selected freely.

$$\underbrace{\lceil d/k \rceil}_{\text{size of } A} + \underbrace{k}_{\text{size of } B} .$$

 Therefore, we can write the total communication as:

$$\text{MAPO total communication} \in \{ k \mid k = 1, 2, \ldots \}.$$

 This is particularly important in communication-efficient FL since viable solutions can be found with
 communication cost $k < d_1$ or $d_1 < k < 2d_1$, which architecture-dependent layer-wise factorization
 can not offer.

