# OpenReview forum: "Reshape-then-Factorize: Communication-Efficient FL via Model-Agnostic Projection Optimization"
_NeurIPS.cc/2025/Conference — Submitted to NeurIPS 2025_

### Official Review · Reviewer_cpPM · 2025-07-02

**Clarity:** 3
**Significance:** 2
**Originality:** 2
**Rating:** 4
**Confidence:** 3

**Summary:**

This submission introduces a new method (MAPO) for reducing the communication overhead of exchanging full model updates in federated learning (FL). The method is based on reshaping and factorizing the full model gradient into a fixed recostruction matrix and trainable projection vector, which makes it independent on per-layer dimensions and architecture.
The method is theoretically analyzed and experiments complement the claims on efficiency.

**Questions:**

1. **About Assumption 3.1:** it is not clear if it is an assumption or if the property of gaussian matrices has been proved. In fact, reading line 155 it seems to be a proven result more than an assumption.
2. **Does the algorithm require full participation?** Similarly, under what conditions do the provided theoretical guarantees hold? From algorithm 1, line 5, it seems that the algorithm requires full client participation, while in theorem 4.3 it is not clear how the number of selected clients affects the rate, e.g. it is not clear the dependency of the noise from the participating clients.
3. **Does Theorem 4.3 assume iid conditions?** Usually FL algortihms need to assume that the variance between clients gradients is bounded (Bounded Gradient Dissimilarity, BGD), which is used to quantify the dependency on heterogeneity. It seems that this assumption is not mentioned in the paper, as assumption 4.2 only bounds the stochastic variance due to mini-batch sampling inside each client. Please clarify if the assumption is implicitly used (i.e. we assume there is zero dissimilarity) or what is the reason the theorem does not depend on it.
4. **Does Theorem 4.3 take into account local steps?** Algorithm 1 line 10 indicates that local epochs are used, but it seems that local steps do not appear in the theorem (e.g. do not influence the choice of the learning rate, and do not appear in the dependency on stochastic noise). Also, why $\eta_t$ remains explicitly reported in the final bound? I would expect to only see the dependency on its upper bound.

**Ethical Concerns:**

["NO or VERY MINOR ethics concerns only"]

**Final Justification:**

The rebutall adeguately addresses most of the concerns, in particular regarding the theory (guarantee under non-iidness and partial participation) and comparison with [2].

The experiments are still on academic dataset, since the added experiments are still too small scale to really represent real-world conditions.

Overall, I lean towards acceptance, but I think the current rating of "borderline accept" reflects well my opinion on the work. Larger-scale experiments with Google Landmarks and/or INaturalist would have been a plus, confirming the effectiveness on setting closer to the real world.

**Limitations:**

yes

**Quality:**

3

**Strengths And Weaknesses:**

## Strenghts
* **The paper in general well-written:** besides being a little imprecise on piece of the theoretical results (see questions), in general the paper is well-written, discusses in sufficient detail the background and related section and is complemented by useful illustrations and tables (e.g. Table 1)
* **The proposed approach is theoretically grounded:** the properties of MAPO are well introduced in section 3.1, which serves as a guide through the theoretical results. There are possibly some typos in the naming of references (e.g. assumptions and definitions are referred as theorems, I believe it is a cleverref config. issue), but overall the organization is clear.
* **Good empirical results:** the provided results support the claims on communication efficiency and accuracy.

## Weaknesses
* **Unclear aspects of Theorem 4.3:** for details please see the questions
* **Unclear postioning of the method regarding FL settings:** This is in part also tackled in the questions. Is the method being proposed for cross-silo or cross-device FL settings?
* **Most of the datasets are academic and small-scale:** I believe the results would be much stronger if larger-scale datasets had been used, like Google Landmarks and INaturalist. A previous paper [1] has introduced federated versions compatible with cross-silo/cross-device settings that authors could use to provide stronger benchmarking.
* **Missing comparison with [2]:** given the similarity in the end objective with this paper, I believe it is necessary to compare the proposed approach with [2], both in the methodological differences (e.g. when one would choose one method or the other in practical cases) and in the experimental results.


[1] Hsu et al., Federated Visual Classification with Real-World Data Distribution, ECCV 2021
[2] Azam et. al, Recycling Model Updates in Federated Learning: Are Gradient Subspaces Low-Rank?, ICLR 2022

---

> ### Author Rebuttal · Authors · 2025-07-30
>
> **Hi Reviewer cpPM:**
>
> We sincerely thank you for your thoughtful and constructive comments. We have addressed all concerns explicitly below and have accordingly updated the manuscript.
>
> ---
> > **Q1: typos**
>
> Thank you for bringing this to our attention. We fixed the issues with cleverref.
>
> ---
> > **Q2: cross-silo or cross-device?**
>
> **R2:** MAPO supports both cross-device and cross-silo FL. The main paper focuses on cross-device with 10% participation, while Appendix B and E include full-participation cross-silo results. Our updated theorem also explicitly supports partial participation.
>
> ---
> > **Q3: The results on larger-scale datasets**
>
> **R3:** Thank you for the suggestion. In addition to academic datasets (MNIST/FMNIST/CIFAR), we included large-scale LEAF [6] tasks (Shakespeare, Sent140). Due to resource limits, we used CelebA with LEAF splits as a practical alternative to [1], offering similar user/data scale with lower cost. Results for cross-silo (10 silos, 20k/silo, full participation) and cross-device (9433 users, ~200/user, 5% participation) are shown below.
>
> |Method|Silo-Acc|Silo-Com|Device-Acc|Device-Com|
> |-|-|-|-|-|
> |FedAvg|91.3|100|88.3|100|
> |FedLoRU|84.9|2.19|77.8|1.37|
> |MAPO|89.6|1.07|86.4|0.98|
>
> Comparison with baselines and plots are placed in Section 6 and Appendix A.
>
> ---
> > **Q4: Missing comparison [2]**
>
> **R4:** We have added LBGM from [2] to our related work and included it in our experiments.
>
> **Methodological Differences**
> 1. **Projection Subspace:**
>    * **LBGM** projects onto a subspace formed by previously collected gradients, requiring several rounds of **full-model communication**.
>    * **MAPO** uses a **random subspace**, avoiding full-model communication entirely.
> 2. **Projection Timing:**
>    * **LBGM** applies projection **after local training**, similar to sketching and EvoFed.
>    * **MAPO** optimizes **within** the projection subspace during local training.
> 3. **Memory Usage:**
>    * **LBGM** stores multiple full-size gradients ($\sim kd$ memory).
>    * **MAPO** only stores a small matrix $\mathbf{A}$ of size $\frac{d}{k}$.
> 4. **Compression Potential:**
>    * LBGM may achieve better compression (smaller $k$) after collecting diverse gradients, but at the cost of higher memory and initial communication, and its effectiveness depends on model architecture [2].
>
> **When to Choose MAPO vs. LBGM**
> * If clients can store multiple gradients and afford some full-model communication, **LBGM or a hybrid strategy** (LBGM + MAPO) can be used: first project onto stored gradients, then onto a random subspace.
> * In realistic FL settings with **limited memory**, **large models**, and **tight bandwidth**, **MAPO is preferable** as it avoids full-model communication and large buffer storage.
>
> |Method|MNIST Acc|MNIST Com|FMNIST Acc|FMNIST Com|CIFAR‑10 Acc|CIFAR‑10 Com|
> |:-:|:-:|:-:|:-:|:-:|:-:|:-:|
> |FedAvg|98.9|100|89.2|100|69.0|100|
> |LBGM|94.8|37.5|84.0|19.2|61.2|28.6|
> |**MAPO**|**98.6**|**2.95**|**88.0**|**3.10**|**68.3**|**1.20**|
>
> Even a few rounds of full-model exchange in LBGM significantly raise the communication cost. Moreover, its reliance on fixed projection directions limits exploration and may lead to **sub-optimal convergence**. In contrast, MAPO enables **fully communication-efficient training** from the start, with **better exploration** via randomized subspaces, making it more suitable for practical FL deployments.
>
> ---
> > **Q5: Assumption 3.1: It is not clear**
>
> **R5:** Thank you for noting this. We restate this assumption as a lemma.
>
> ---
> **Q6: Theorem 4.3 Assumptions**
>
> **R6:** We thank the reviewer for their close attention to the theory. The three key concerns relate to **(i)** non-iid assumptions, **(ii)** number of local steps, and **(iii)** partial client participation.
>
> We address each point step by step, culminating in a generalized convergence theorem that explicitly incorporates all three. These refinements do not change our methodology or experiments, but rather extend the theory while remaining fully compatible with the original assumptions.
>
> ---
> > **Q6.1: Does Theorem 4.3 assume iid conditions?**
>
> **R6.1: (i) non-iid assumption**
>
> You are correct that our initial statement implicitly assumed IID conditions. To make this explicit and to generalize our result, we added a standard BGD assumption (now Assumption 4.3):
>
> $$\frac{1}{M}\sum_{j=1}^M\|\nabla\mathcal{L}^j(W)-\nabla\mathcal{L}(W)\|^2 \leq \sigma_g^2,\quad\forall W.$$
>
> We have carefully revised the proof of Theorem 4.3, making the dependence on client heterogeneity explicit:
>
> $$\frac{1}{4H_T}\sum_{t=0}^{T-1}\eta_t\mathbb{E}\|\nabla\mathcal{L}(W^t)\|^2\leq\frac{\mathbb{E}[\mathcal{L}(W^0)]-\mathcal{L}^*}{H_T}+2(\epsilon+\beta+\beta\epsilon)\,\left(\sigma_l^2+\sigma_g^2\right)\frac{1}{H_T}\sum_{t=0}^{T-1}\eta_t^2.$$
>
> This generalizes our theoretical analysis; the IID case is recovered by setting $\sigma\_g^2 = 0$. This adjustment follows standard FL practice \[2] and does not affect our experiments, main claims, or the $\eta\_t$ upper bound. The non-IID version will be added to Section 4 of the revised manuscript.
>
> ---
> > **Q6.2: Does Theorem 4.3 take into account local steps?**
>
> **R6.2: (ii) number of local steps**
>
> Thank you for pointing this out. The confusion arises from inconsistent notation: Algorithm 1 uses \$\eta\$ as the per-step learning rate for $\nabla B$, while Theorem 4.3 uses \$\eta\_t\$ to denote the total update across $E$ local steps ($W^{t+1} = W^t - \eta\_t\bar{B}^tA^t$). We clarify this by using \$\eta\_t\$ consistently and making $E$ explicit in Theorem 4.3. The revised theorem is:
>
> $$\frac{1}{4H_T}\sum_{t=0}^{T-1}\eta_t
> \mathbb{E}\bigl[|\nabla\mathcal{L}(W^t)|^2\bigr] \le \frac{\mathbb{E}\bigl[\mathcal{L}(W^0)\bigr]-\mathcal{L}^*}{EH_T} +
> 2E\bigl(\epsilon+\beta+\beta\epsilon\bigr)(\sigma_l^2+\sigma_g^2)
> \frac{1}{H_T}\sum_{t=0}^{T-1}\eta_t^2$$
> The upper-bound condition on $\eta_t$ is unchanged.
> Increasing the number of local epochs $(E\uparrow)$ shrinks the bias term (first RHS term) but linearly amplifies stochastic noise, reflecting the classical communication–variance trade-off.
> Similarly, given $E=1$, we reach the previous theorem.
>
> ---
> > **Q6.3: Does the algorithm require full participation?**
>
> **R6.3: (iii) partial client participations**
>
> The algorithm and theorem are written under the assumption of implicit full client participation; however, this is not a requirement. Our methodology can also be applied to partial participation. The main paper uses a 10% participation rate, while the appendix includes full participation experiments. We clarify this in the updated manuscript as follows:
>
> We now state the sampling scheme explicitly as **Assumption 4.4**.
>
> At every communication round $t$:
> Select a subset $S_t \subset [M]$ with $|S_t| = m < M$ **uniformly at random without replacement**. The server aggregates projected gradients via:
> $$\bar B^t=\frac{1}{m}\sum_{j\in S_t} B^{t,j}.$$
>
> To keep the bound compact, the sampling variance is absorbed into one constant:
> $$\sigma_{\text{het}}^2 = \sigma_g^{2}\Bigl(1 + \rho\Bigr), \qquad \rho = \frac{M - m}{m\bigl(M - 1\bigr)}$$
> -   **Full participation** ($m = M$): $\rho = 0$ and $\sigma_{\text{het}}^2 = \sigma_g^2$.
> -   **Low participation** ($m \ll M$): $\sigma_{\text{het}}^2 \approx \sigma_g^2/m.$
>
> Therefore, we obtain the following theorem:
> $$\frac{1}{4H_T}\sum_{t = 0}^{T-1}\eta_t \mathbb{E}\left[\bigl\lVert\nabla\mathcal L\left(W^{t}\right)\bigr\rVert^{2}\right] \le \frac{\mathbb{E}\left[\mathcal L\left(W^{0}\right)\right] - \mathcal L^{*}}{EH_T} + 2E\bigl(\epsilon + \beta + \beta\epsilon\bigr) \bigl(\sigma_{l}^{2} + \sigma_{\text{het}}^{2}\bigr)\, \frac{1}{H_T}\sum_{t = 0}^{T-1}\eta_t^{2},$$
>
> Or to have participation explicitly in the theorem, we can write it as:
> we obtain
> $$\frac{1}{4H_T}\sum_{t = 0}^{T-1}\eta_t\mathbb{E}\left[\bigl\lVert\nabla\mathcal L\left(W^{t}\right)\bigr\rVert^{2}\right] \le \frac{\mathbb{E}\left[\mathcal L\left(W^{0}\right)\right] - \mathcal L^{*}}{EH_T}+2E\bigl(\epsilon + \beta + \beta\epsilon\bigr) \bigl(\sigma_{l}^{2} + \sigma_{g}^{2} + \sigma_g^2(\frac{M - m}{m(M - 1)})\bigr)\, \frac{1}{H_T}\sum_{t = 0}^{T-1}\eta_t^{2},$$
>
> **Interpretation**
> - The **bias term** is unchanged.
> - The **variance term** now contains $\sigma_{l}^{2}$ (mini-batch noise) and $\sigma_{\text{het}}^{2}$ (heterogeneity **plus** sampling noise).
> - Increasing the number of active clients $m$ reduces $\rho$ and thus $\sigma_{\text{het}}^{2}$, recovering the earlier non-IID bound when $m=M$.
> - This structure mirrors the partial participation assumption in Theorem 3 of [3] for FedAvg, with additional JL-projection factors.
>
> ---
> > **Q7: Why $\eta_t$ remains explicitly?**
>
> **R7:**
>
> **(1) Keeping \$\eta\_t\$ explicit is standard in non-convex FL theory.** (see Theorem 4.10 in [4] and Theorem 4.3 in [5])
> Particularly, Theorem 4.10 in [4] presents the bound with the full stepsize sequence \$\alpha\_k\$:
> $$
> \sum\_{k=1}^{K} \alpha_k\, \mathbb{E}[\|\nabla F(w_k)\|_2^{2}] \le  \frac{2(\mathbb{E}[F(w\_1)] - F\_{\inf})}{\mu} + \frac{LM}{\mu} \sum\_{k=1}^{K} \alpha_k^{2}
> $$
>
> **(2) Convergence to a stationary point.**
> If the Robbins–Monro conditions hold (\$\sum\_t \eta\_t = \infty\$, \$\sum\_t \eta\_t^2 < \infty\$), then \$\lim\_{T \to \infty} \min\_{t\<T} \mathbb{E}\[|\nabla \mathcal{L}(W^t)|^2] = 0\$, as in Theorem 4.10 of \[4] and Theorem 4.3 of [5]. Our proof follows the same steps, and we will clarify this explicitly.
>
> We hope this resolves the concern and demonstrates that our analysis is consistent with, and no weaker than, prior work, while offering a more general and informative bound. Thank you again for your insightful feedback.
>
> ---
> ```
> [3] Li et al. On the convergence of fedavg on non-iid data.
> [4] Bottou et al. Optimization methods for large-scale machine learning.
> [5] Kim et al. Achieving lossless gradient sparsification via mapping to alternative space in federated learning.
> [6] Caldas et al. Leaf: A benchmark for federated settings.
> ```

---

> > ### Comment · Reviewer_cpPM · 2025-08-05
> >
> > I thanks the authors for the rebuttal, in particular for integrating the theory to handle non-iidness and partial participation and for additional comparison with LBGM. The rebutall adeguately addresses most of the concerns, and I lean towards acceptance. Larger-scale experiments with Google Landamarks and/or INaturalist would have been a plus, confirming the effectiveness on setting closer to the real world.

---

### Official Review · Reviewer_TUEL · 2025-07-02

**Clarity:** 3
**Significance:** 3
**Originality:** 4
**Rating:** 5
**Confidence:** 3

**Summary:**

This paper focuses on communication-efficient federated learning and aims to improve the flexibility of low-rank decomposition methods. It proposes reshaping and factorizing the gradient via a universal vector. The proposed method is evaluated on several datasets and demonstrates improved performance.

**Questions:**

- Are there any other factors that could affect the reinitialization of A? Is sharing the seed sufficient?

**Ethical Concerns:**

["NO or VERY MINOR ethics concerns only"]

**Final Justification:**

Thanks for the authors detailed response, I have no more questions.

**Limitations:**

yes

**Quality:**

4

**Strengths And Weaknesses:**

Strengths
- The topic is relevant and important in federated learning.
- The motivation for improving low-rank decomposition is well demonstrated.
- The method is novel, and the experiments show good results.

Weaknesses
- The practical implications of the convergence analysis are unclear and could be further elaborated.
- The comparison methods for quantization are somewhat outdated; it would be better to include more recent quantization works in the comparison.
- The advantages of using fresh A are obvious, but it would be better to discuss the potential reasons further.

---

> ### Author Rebuttal · Authors · 2025-07-30
>
> **Hi Reviewer TUEL:**
>
> We sincerely thank you for your thoughtful and constructive comments. We have addressed all concerns explicitly below and have accordingly updated the manuscript.
>
> ---
> > **Q1: The practical implications of the convergence analysis are unclear and could be further elaborated.**
>
> **R1:** Thank you for raising this concern we clarfied our theorem practical implications as by elaborating on its relation with FedAvg and compression rate as follows:
>
> * **When the reconstruction error is zero** (\$\varepsilon = 0\$, i.e., \$k = d\$), our bound collapses exactly to FedAvg’s tightest result of \$\mathcal{O}(1/\sqrt{T})\$ (McMahan et al., 2017).
> * **Under compression** (\$\varepsilon > 0\$), the only difference is a *constant* multiplicative factor \$(\varepsilon + \beta + \varepsilon\beta)\$ that reflects the projection bias (**no slower rate is introduced**).
>
> Formally, for step sizes satisfying \$\eta\_t \leq \bar{\eta}\$, we prove:
> $$
> \frac{\sum_{t=0}^{T-1} \eta_t\,\mathbb{E}[\|\nabla \mathcal{L}(W^t)\|^2]}
>      {\sum_{t=0}^{T-1} \eta_t}
> = \mathcal{O}(T^{-1/2}) \times (\varepsilon + \beta + \varepsilon\beta),
> $$
> which ensures that the expected gradient norm vanishes at the rate of \$T^{-1/2}\$. In non-convex optimization, this guarantees convergence to a **stationary point**, just as FedAvg does. Similar formulations can be found in [1] (see Theorem 4.10) and [2] (see Theorem 4.3).
>
> Practical implication on the **Projection Rank \$k\$ (traffic per round):** Since \$\varepsilon \propto k^{-1/2}\$, reducing traffic by \$100\times\$ (e.g., sending only 1% of parameters) increases the multiplicative constant by roughly \$10\times\$. In practice, the empirical accuracy drop in experiments is around 1%.
>
> ---
> > **Q2: The comparison methods for quantization are somewhat outdated; it would be beneficial to include more recent works on quantization in the comparison.**
>
> **R2:** We conducted additional experiments using **FedFQ** \[3], a more recent quantization baseline that employs fine-grained quantization per parameter. The results are presented below alongside the relevant baselines from our paper:
>
> |Method|CIFAR‑10 Acc(%)|CIFAR‑10 Com.|CIFAR‑100 Acc(%)|CIFAR‑100 Com.|Shakespeare Acc(%)|Shakespeare Com.|
> |---|---|---|---|---|---|---|
> |**FedAvg**|69.0|100|43.47|100|41.86|100|
> |**Quant**|*67.40*|15.2|40.05|6.10|35.45|10.11|
> |**FedFQ**|66.01|*3.64*|*40.11*|*3.71*|*39.28*|*1.70*|
> |**MAPO**|**68.30**|**1.20**|**40.16**|**0.91**|**39.96**|**0.13**|
>
> As the results show, **MAPO's low-rank projection** still provides a clear advantage in **communication efficiency** compared to quantization-based methods.
>
> We would like to emphasize that **sparsification** and **quantization** methods (e.g., FedPAQ, FedFQ) are universal and **orthogonal** to low-rank gradient projection. Therefore, they can be **combined with MAPO** to achieve even greater compression.
>
> In preliminary experiments, we found that applying a lightweight sparsification (≈ 30%) followed by quantization (e.g., reducing from 32 bits to 16–12 bits) **significantly improves communication efficiency**, while only **marginally affecting accuracy**.
>
> However, the results reported in our paper are based on **standalone MAPO**, without any hybrid enhancements, in order to isolate and highlight the core performance of our projection method.
>
> ---
> > **Q3: The advantages of using fresh A are obvious, but it would be better to discuss the potential reasons further.**
>
> **R3:** Refreshing the reconstruction matrix $A$ each communication round helps for four main reasons:
> 1.  **Prevention of rank‑depletion.** A fixed low‑rank basis can exhaust useful descent directions over time; resampling $A$ guarantees exploration of new, approximately orthogonal subspaces and keeps gradient information “alive.”
>
> 2.  **Implicit regularization.** A fresh random basis injects isotropic noise, acting much like DropConnect on the gradient. This reduces client-specific overfitting and consistently narrows the training–test gap. We observed that while most baselines required L1 and/or L2 regularization to stabilize and avoid overfitting, MAPO could rely solely on this implicit regularization.
>
> 3.  **Coverage of minority features.** Multiple random subspaces increase the probability that minority‑class directions receive non‑zero projection (per the Johnson–Lindenstrauss lemma, intersection probability $\ge 1-\delta$).
>
> 4.  **Synergy with compression.** Because the projection error becomes i.i.d. across rounds, its entropy drops, which simplifies any subsequent quantization or entropy‑coding to be applied after the MAPO projection $B$.
>
> We have added this discussion and a new ablation to the Appendix, which describes these trends.
>
> ---
> > **Q4: Are there any other factors that could affect the re‑initialization of A?**
>
> **R4:** In principle, $A$ may be drawn from a Gaussian family $\mathcal{N}(\mu_t,\sigma_t^2 I)$. In the current work, we fix $\mu_t=0$ (unbiased) and $\sigma_t=\sigma_0$ (constant), so the _seed_ is the only transmitted scalar. Two lightweight extensions could further tailor the sampling:
>
> -   **Annealed variance:** set $\sigma_t=\sigma_0/\sqrt{t}$ so early rounds explore widely while later rounds refine.
>
> -   **Momentum‑aligned mean:** choose $\mu_t\propto m_t$, where $m_t$ is the global momentum, nudging the random basis toward historically useful directions.
>
> Either variant would add at most one scalar value or none per round (depending on the implementation), which is negligible overhead relative to the projection vector length $k$.
>
> ---
> > **Q5: Is sharing the seed sufficient?**
>
> **R5:** Yes, for the design used in our experiments, _one common seed per round_ completely determines $A$ because $\mu=0$ and $\sigma$ is constant.
>
> If future variants employ adaptive $\mu_t$ or $\sigma_t$, an additional parameter might accompany the seed.
>
> ---
> ```
> [1] Bottou, L., Curtis, F. E., & Nocedal, J. (2018). Optimization methods for large-scale machine learning. _SIAM review_, _60_(2), 223-311.
> [2]   Kim, D. Y., Han, D. J., Seo, J., & Moon, J. (2024, July). Achieving lossless gradient sparsification via mapping to alternative space in federated learning. In _Forty-first International Conference on Machine Learning_
> [3] Li, H., Xie, W., Ye, H., Ma, J., Ma, S., & Li, Y. (2024). FedFQ: Federated Learning with Fine-Grained Quantization. _arXiv preprint arXiv:2408.08977_.
> ```

---

### Official Review · Reviewer_PnwW · 2025-07-03

**Clarity:** 3
**Significance:** 3
**Originality:** 3
**Rating:** 4
**Confidence:** 4

**Summary:**

MAPO is a model agnostic method of compressing gradients for efficient communication between client and server. The merits of MAPO are demonstrated through theoretical analysis and practical application.

**Questions:**

* as tasks become more complex/ have higher dimensional data, it seems likely that the minimum value of k required to get a good approximation would increase. Have you tried experiments with high dimensional data? (medical images?), is the k required to train on this data prohibitive?
* re-defining A could help with subspace exploration but it could also make updates unstable with gradients that change drastically between updates, particularly early in training. Have you observed this behaviour? Are there steps taken to mitigate it?

**Ethical Concerns:**

["NO or VERY MINOR ethics concerns only"]

**Final Justification:**

The authors did a thorough job of addressing concerns for the rebuttal. I recommend acceptance but will keep my original score of 4 since I think that better reflects my impression of the overall contribution.

**Limitations:**

yes

**Quality:**

3

**Strengths And Weaknesses:**

**Strengths**
* The authors do a good job of comparing MAPO to CEFL methods and highlighting its advantages over existing methods
* Model agnostic
* Strong set of baselines that demonstrate several important use cases

**Weaknesses**
* No explicit evaluation of the ability of the low rank projection to capture meaningful gradient directions. This evaluation would lend strength to the claim that MAPO's performance is due to dynamic subspace exploration
* may overlook important minority features for clients with long tailed distributions

---

> ### Author Rebuttal · Authors · 2025-07-30
>
> **Hi Reviewer PnwW:**
>
> We sincerely thank you for your thoughtful and constructive comments. We have addressed all concerns explicitly below and have accordingly updated the manuscript.
>
> ---
> > **Q1: No explicit evaluation of the ability of the low rank projection to capture meaningful gradient directions.**
>
> **R1:** Thank you for raising this point. We define a meaningful low-rank projection as one that closely approximates the full gradient, not just reduces loss. To test whether dynamic subspace exploration improves this, we compared fixed vs. dynamic projections on MNIST and FMNIST over 500 FL rounds.
>
> The cosine similarity was near zero for both, which is expected due to the near-orthogonality in high dimensions. However, **L2 distance** was consistently **lower** for dynamic projection, indicating a closer approximation to the proper gradient.
>
> The results of the cumulative L2 distances are shown in the table below.
>
> **Cumulative L2 Distance (FMNIST)**
> |Method|0|100|200|300|400|500|
> |-|-|-|-|-|-|-|
> |**Frozen**|38.0|46.5|50.8|53.8|55.2|55.5|
> |**Fresh**|38.0|44.1|47.0|50.2|52.5|53.3|
>
> **Cumulative L2 Distance (MNIST)**
> |Method|0|100|200|300|400|500|
> |-|-|-|-|-|-|-|
> |**Frozen**|30.0|47.0|52.7|57.8|61.5|64.0|
> |**Fresh**|30.0|42.5|46.8|50.8|53.8|56.2|
>
> We have included a comprehensive ablation study in this regard in Appendix J.
>
> ---
> > **Q2: may overlook important minority features for clients with long-tailed distributions**
>
> **R2:** Thank you for raising this important point. MAPO’s projection is *orthogonal* to common data imbalance remedies in FL. In practice, one can:
>
> **(i)** Apply class reweighting or logit adjustment *locally, before the projection* \[1, 2];
>
> **(ii)** Combine MAPO with heterogeneity-robust optimizers or data selection strategies that operate *prior to projection*, such as **FedProx** \[3] or **FedBalancer** \[4], ensuring that minority-class signals are preserved;
>
> **(iii)** Increase either the projection rank \$k\$ or the number of local epochs, both of which expand the effective subspace available to represent minority features.
>
> We will incorporate these clarifications as a Remark in Section 3 of the manuscript.
>
> ---
> > **Q3: as tasks become more complex/ have higher dimensional data, it seems likely that the minimum value of k required to get a good approximation would increase.**
>
> **R3:** Thank you so much for raising this concern. You are right, increasing the task complexity and higher input dimensionality necessitate a larger projection dimension \$k\$ and a greater number of model parameters.
>
> To evaluate this, we conducted additional experiments on the **HAM10000** medical imaging dataset for skin cancer detection. We tested both low-resolution (\$28 \times 28 \times 3\$) and high-resolution (\$224 \times 224 \times 3\$) inputs using **CNN** and **WideResNet (WRN)** architectures. The results are summarized in the table below, with columns indicating the model architecture, number of parameters, input dimensions, and the corresponding projection dimension \$k\$ used.
>
> |Method|CNN-3M 28×28 k=2¹²|CNN-3M 28×28 k=2¹²|CNN-208M 224×224 k=2¹⁸|CNN-208M 224×224 k=2¹⁸|WRN-2.8M 28×28 k=2¹²|WRN-2.8M 28×28 k=2¹²|WRN-2.8M 224×224 k=2¹⁶|WRN-2.8M 224×224 k=2¹⁶|
> |:-:|:-:|:-:|:-:|:-:|:-:|:-:|:-:|:-:|
> ||**Acc.**|**Com.**|**Acc.**|**Com.**|**Acc.**|**Com.**|**Acc.**|**Com.**|
> |FedAvg|77.05|100.00|79.76|100.00|79.13|100.00|81.83|100.00|
> |Sparse|71.47|1.67|74.09|2.07|73.57|1.72|79.07|13.72|
> |Quant|75.51|10.31|78.23|14.21|77.82|5.63|79.32|28.44|
> |EvoFed|71.82|4.57|74.13|3.83|73.97|10.81|78.11|18.72|
> |FedLoRU|74.60|1.33|78.18|1.51|77.01|1.52|78.98|12.43|
> |**MAPO**|**76.58**|**1.07**|**79.20**|**1.23**|**78.20**|**0.93**|**80.16**|**9.27**|
>
> As observed, gradients generated from more complex tasks exhibit higher information content and entropy, making them more resilient to compression for all baselines. Nevertheless, **MAPO consistently outperforms other baseline methods**, even when a higher projection dimension is required.
>
> We included these new findings in Appendix A.
>
> ---
> > **Q4: is the k required to train on this data prohibitive?**
>
> **R4:** We address this question from both computational and communication perspectives.
>
> **Computation and Memory:**
> It is important to note that a *larger* value of \$k\$ actually *reduces* the memory and computation requirements of MAPO (as shown in **Proposition 3.6**). This is because the size of the reconstruction matrix \$\mathbf{A}\$ decreases as \$k\$ increases. Therefore, MAPO not only avoids computational or memory bottlenecks but also improves memory efficiency by requiring a smaller \$\mathbf{A}\$ matrix.
>
> **Communication Bandwidth:**
> On the other hand, a *larger* value of \$k\$ increases communication bandwidth requirements and reduces the compression rate. However, our experiments show that the **communication efficiency** of MAPO (defined as the *minimum communication required to reach a target accuracy*) remains superior to baseline methods.
>
> We find that more complex tasks and higher input dimensionality tend to produce more expressive gradients, which are inherently harder to compress across all methods.
>
> Notably, we observe that gradient signals from **fine-tuning tasks** are much easier to compress at high compression rates. For instance, while the appropriate \$k\$ for MAPO during **pre-training** is approximately \$100 \sim 300\times\$ smaller than the total number of parameters, in **fine-tuning** scenarios it can reach compression ratios as high as \$1000 \sim 10000\times\$.
>
> **Practical Suggestion:**
> To reduce communication costs in FL for challenging tasks or high-dimensional data, a **practical approach** is to start from a **pre-trained model** and perform **fine-tuning** in the federated setting. This leverages both better initialization and higher compressibility of gradients during fine-tuning.
>
> ---
> > **Q5: re-defining A could help with subspace exploration but it could also make updates unstable**
>
> **R5:** Thank you for raising this question. The behavior we observed aligns exactly with your comment.
>
> However, for most tasks, the advantage of using a fixed \$\mathbf{A}\$ is **marginal** in the early rounds and diminishes quickly as training progresses. Results for Sentiment140, Shakespeare, tinyImageNet, and CIFAR-100 are shown in the tables below.
> We have included the following results in Appendix J.
>
> **Sent140 (100 clients)**
> | Method|1|100|200|300|400|500| 600 | 700 | 800 | 900 | 1000 | 1100 | 1200 | 1300 | 1400 |
> |-|-|-|-|-|-|-|-|-|-|-|-|-|-|-|-|
> | Fresh-A | 62.72% | 62.72% | 62.72% | 63.03% | 64.18% | **66.61%** | **67.09%** | **69.59%** | **69.78%** | **70.14%** | **72.31%** | **72.88%** | **73.58%** | **73.92%** | **74.76%** |
> | Frozen-A   | **62.73%** | **62.75%** | **63.37%** | **64.88%** | **64.69%** | 65.63% | 66.27% | 66.83% | 66.32% | 66.03% | 66.85% | 66.47% | 66.73% | 66.78% | 67.09% |
> | *Diff* | *-0.01%* | *-0.03%* | *‑0.65%* | *‑1.85%* | *‑0.50%* | *0.98%* | *0.82%* | *2.76%* | *3.46%* | *4.11%* | *5.46%* | *6.41%* | *6.85%* | *7.14%* | *7.67%* |
>
>
> ---
> **Shakespeare**
> | Method | 1 | 50 | 100 | 150 | 200 | 250 | 300 | 350 | 400 | 450 | 500 | 550 | 600 | 650 | 700 |
> |-|-|-|-|-|-|-|-|-|-|-|-|-|-|-|-|
> | Fresh-A|14.67%|15.33%|22.18%| **28.33%** | **30.60%** | **32.71%** | **33.79%** | **34.99%** | **35.86%** | **36.54%** | **36.75%** | **37.33%** | **37.82%** | **38.36%** | **38.63%** |
> | Frozen-A   | **14.69%** | **20.47%** | **23.73%** | 25.40% | 25.99% | 26.71% | 26.60% | 27.13% | 27.67% | 28.13% | 28.04% | 28.27% | 27.96% | 28.32% | 28.18% |
> | *Diff* | *-0.02%* | *‑5.14%* | *‑1.55%* | *2.93%* | *4.61%* | *6.00%* | *7.19%* | *7.86%* | *8.19%* | *8.41%* | *8.71%* | *9.06%* | *9.86%* | *10.04%* | *10.45%* |
>
> ---
> **tinyImageNet**
> | Method | 1 | 10 | 20 | 30 | 40 | 50 | 60 | 70 | 80 | 90 | 100 | 110 | 120 | 130 | 140 |
> |-|-|-|-|-|-|-|-|-|-|-|-|-|-|-|-|
> | Fresh-A | 0.74% | 1.37% | 2.22% | 4.90% | 6.95% | **8.55%** | **11.37%** | **13.52%** | **15.24%** | **17.76%** | **20.05%** | **21.84%** | **23.00%** | **24.11%** | **24.79%** |
> | Frozen-A | **0.75%** | **1.43%** | **4.00%** | **5.77%** | **7.25%** | 8.03% | 9.22% | 10.01% | 10.66% | 10.52% | 10.40% | 10.23% | 9.94% | 9.82% | 9.69% |
> | *Diff* | *-0.01%* | *‑0.06%* | *‑1.78%* | *‑0.87%* | *‑0.30%* | *0.52%* | *2.15%* | *3.51%* | *4.58%* | *7.24%* | *9.65%* | *11.61%* | *13.07%* | *14.29%* | *15.10%* |
>
> ---
> **CIFAR‑100**
> | Method | 1 | 10 | 20 | 30 | 40 | 50 | 60 | 70 | 80 | 90 | 100 | 110 | 120 | 130 | 140 |
> |-|-|-|-|-|-|-|-|-|-|-|-|-|-|-|-|
> | Fresh-A | 1.19% | 8.01% | **14.44%** | **18.11%** | **21.79%** | **23.12%** | **25.49%** | **27.51%** | **28.23%** | **30.60%** | **32.18%** | **32.30%** | **31.41%** | **33.97%** | **34.01%** |
> | Frozen-A | **1.55%** | **8.79%** | 12.83% | 14.33% | 15.32% | 15.69% | 15.45% | 16.58% | 17.02% | 16.63% | 17.67% | 17.92% | 18.08% | 18.32% | 18.55% |
> | *Diff* | *‑0.36%* | *‑0.78%* | *1.61%* | *3.78%* | *6.47%* | *7.43%* | *10.04%* | *10.93%* | *11.21%* | *13.97%* | *14.51%* | *14.38%* | *13.33%* | *15.65%* | *15.46%* |
>
> ---
> > **Q6: "Are there steps taken to mitigate it?"**
>
> **R6:** This question reflects the trade-off between **exploration** and **exploitation**. For simplicity, we adopt *continuous exploration* by redefining \$\mathbf{A}\$ each round.
>
> However, the update frequency can be adjusted (e.g., updating \$\mathbf{A}\$ every few rounds or using a fixed \$\mathbf{A}\$ for warm-up, then switching to exploration) to balance convergence and simplicity.
>
> The **number of local steps** also plays a role: with enough local steps, clients can effectively exploit the current subspace before exploring a new one.
>
> ---
> ```
> [1] Menon et al. Long-tail learning via logit adjustment.
> [2] Cui et al. Class-balanced loss based on effective number of samples.
> [3] Li et al. Federated optimization in heterogeneous networks.
> [4] Shin et al. Fedbalancer: Data and pace control for efficient federated learning on heterogeneous clients.
> ```

---

> > ### Comment · Reviewer_PnwW · 2025-08-05
> >
> > Thank you for your rebuttal, I appreciate the additional experiments provided to address my comments and concerns. I am leaning towards accept but still think the potential instability of $\mathbf{A}$ in early training is a concern, particularly as the scale and complexity of the data increases. It would have been nice to see the behaviour of MAPO under these conditions.

---

> ### Author Response · Authors · 2025-08-09
>
> We appreciate the reviewer’s acknowledgment of our rebuttal and their concerns regarding training stability when reintroducing $A$. Below, we provide additional clarifications and results.
>
> ---
>
> **First**, the learning curves in Figure 5 and Appendix A already illustrate the stability of MAPO training, even in the early rounds, across seven datasets and tasks of varying sizes. Additionally, the tables in the rebuttal answer **R5** show that **Fresh-A** remains stable, with only a slight drop in accuracy compared to **Frozen-A** in a few early rounds, and ultimately converges to a higher accuracy. In contrast, **Frozen-A** tends to get stuck in a local minimum.
>
> **Second**, we emphasize that reinitialization of $A$ occurs only **once per FL round**, which itself contains many local training steps (multiple batches or epochs). This means training proceeds in a fixed subspace for many steps, and the subspace is updated only after server aggregation. Updating $A$ at every local step (per batch) would indeed risk significant instability in early training, as the reviewer noted, but this is **not** the case with MAPO.
>
> **Lastly**, to further evaluate stability as data scale and complexity increase, we provide results on the HAM10000 and CelebA datasets. The tables below compare **Fresh-A** (MAPO) with **Frozen-A** and two additional variants:
>
> * **Frozen-First-50**: $A$ is frozen for the first 50 rounds only.
> * **Semi-Fresh-A**: $A$ is updated every two rounds.
>
> These results confirm that MAPO maintains stability and competitive performance across settings.
>
> ---
>
> **Celeb A**
> |Method|0|10|20|30|40|50|60|70|80|90|100|
> |:--|--:|--:|--:|--:|--:|--:|--:|--:|--:|--:|--:|
> |Fresh-A            |49.97|57.23|69.83|86.04|89.57|90.29|91.06|91.19|91.28|91.31|91.32|
> |Semi-Fresh-A   |49.97|54.67|60.52|63.01|70.13|80.61|84.14|85.79|85.89|87.42|88.06|
> |Frozen-Frist-50|49.97|50.01|50.07|53.42|56.13|57.81|61.56|73.22|83.96|86.84|88.54|
> |Frozen-A          |49.97|50.01|50.07|53.42|56.13|57.81|59.51|60.72|61.23|61.81|62.25|
>
> ---
>
> **HAM10000**
> |Method|0|10|20|30|40|50|60|70|80|90|100|
> |:--|--:|--:|--:|--:|--:|--:|--:|--:|--:|--:|--:|
> |Fresh-A            |67.99|68.13|69.08|70.41|70.68|71.32|72.26|72.75|73.17|73.79|74.13|
> |Semi-Fresh-A   |67.99|68.08|68.44|69.77|69.86|69.54|70.44|70.75|71.03|71.12|70.96|
> |Frozen-Frist-50|67.99|67.99|67.99|67.99|67.99|67.99|68.21|69.38|70.07|71.04|72.97|
> |Frozen-A          |67.99|67.99|67.99|67.99|67.99|67.99|67.99|67.99|67.99|67.99|67.99|
>
> ---
>
> As shown in the tables, with larger and more complex datasets, a fixed small subspace cannot carry sufficient information for effective training. In such cases, higher exploration, achieved by introducing a new subspace each round, proves more beneficial for stable and successful convergence.
>
>
> We sincerely thank the reviewer for their valuable feedback and the opportunity to clarify these points.

---

### Official Review · Reviewer_b6Vz · 2025-07-04

**Clarity:** 3
**Significance:** 4
**Originality:** 4
**Rating:** 5
**Confidence:** 4

**Summary:**

To reduce the communication overhead in federated learning, the authors proposed a model-agnostic projection optimization method based on low-rank decomposition technology, which reshapes and factorizes the gradient of the complete model into a fixed reconstruction matrix and a trainable projection vector, while reducing the communication overhead and solving the limitations of low-rank decomposition technology. Finally, experiments have proved the effectiveness of MAPO in a variety of federated learning scenarios.

**Questions:**

(1). There are some symbolic errors in the text and figures, such as "△W" in "Multi-Layer Gradient Decomposition" in Figure 3, and some parameters in Table 4 lacking units. The reviewer believes that the description and symbols in the text and figures, tables, and algorithms need to be consistent. If the author can analyze this problem in depth and make a comprehensive check and correction, it will make the research more comprehensive and further enhance the authority of the article.
(2). In the paper, MAPO requires each client to use the gradient descent algorithm to update the projection, and the server and client update the model based on the aggregated projection. The reviewer is curious that the author did not mention the threat model of the scheme in the paper. If the client maliciously tampers with the corresponding projection vector, will it cause the final model to be unable to be reconstructed? What impact will it have if the server broadcasts the wrong projection vector?  The reviewer hopes that the author can explain it.
(3). The labels of Figures 5, 6, and 8 in the article cover up some curves, affecting the readers' observation of the experimental results. The reviewer hopes that the author can adjust the above figures to improve the readability of the article.

**Ethical Concerns:**

["NO or VERY MINOR ethics concerns only"]

**Final Justification:**

All the concerns have been addressed. I maintain  the original score.

**Limitations:**

yes

**Quality:**

4

**Strengths And Weaknesses:**

(1)In terms of quality, the submitted paper is reasonable in its technical application, and the designed technical solutions and their performance in specific scenarios are fully demonstrated through rigorous convergence analysis, theoretical proof and systematic experimental verification.The research method of the paper is appropriate, the experimental design is perfect, and the whole paper is a well-structured and detailed academic paper. It is worth affirming that the author not only clearly explains the academic contribution of this study, but also objectively analyzes the limitations of the current solution in the discussion section, which reflects the rigor of the solution.
(2)In terms of clarity, the content of the submitted paper is clearly stated, the article structure is well-organized, and the layout is standardized and reasonable, which can effectively guide readers to understand the innovative points and core contributions of the research.
(3)In terms of significance, this paper has important practical application value for federated learning, and provides a foundation for subsequent research on low-overhead, high-performance, and deployable federated learning. At the same time, the paper is superior to traditional efficient communication solutions in terms of communication overhead and flexibility. In addition, the author verifies the applicability of the method in practical environments through overhead and performance analysis under multiple datasets and heterogeneous data.
(4)In terms of originality, this paper starts with the balanced optimization of communication overhead, model performance, and computational flexibility. Compared with previous studies, it more comprehensively considers the coordinated optimization of computational flexibility and model performance on the basis of reducing communication overhead, and cites relevant literature to support its innovation. The optimized federated learning communication strategy proposed in the study takes into account flexibility and high performance while reducing overhead, and verifies its applicability through heterogeneous data environments, providing new optimization ideas for the development of the field.

---

> ### Author Rebuttal · Authors · 2025-07-30
>
> **Hi Reviewer b6Vz:**
>
> We sincerely thank you for your thoughtful and constructive comments. We have addressed all concerns explicitly below and have accordingly updated the manuscript.
>
> ---
> > **Q1:** There are some symbolic errors in the text and figures, such as "△W" in "Multi-Layer Gradient Decomposition" in Figure 3, and some parameters in Table 4 lacking units.
>
> **R1:** Thank you for pointing this out. We will comprehensively revise and ensure consistency in all symbols and units across figures, tables, and algorithms to ensure accuracy and clarity.
>
> Specifically, we have revised the usage of \$\Delta W\$ and ensured the appropriate use of \$\Delta W'\$ wherever necessary.
>
> Regarding Table 4, it currently reports the number of parameters and does not reflect the actual number of bits per parameter. The units shown (**M** = million, **K** = thousand) represent the **number of parameters**, not the **memory size**. We will update these values to use **GB** and **MB** units, assuming **32bits** per parameter, for a more accurate representation.
>
> ---
> > **Q2:** The reviewer is curious that the author did not mention the threat model of the scheme in the paper. If the client maliciously tampers with the corresponding projection vector, will it cause the final model to be unable to be reconstructed?
>
> > What impact will it have if the server broadcasts the wrong projection vector?
>
> **R2:** Thank you for the question. We now explicitly clarify our threat model as follows: MAPO assumes a standard FL scenario with honest-but-curious servers and honest clients.
>
> Malicious tampering with the projection vector $\mathbf{B}$ is algebraically equivalent to gradient poisoning; therefore, Byzantine-robust aggregation methods [1, 2, 3] are directly applicable.
>
> Server-side errors resemble standard FL model update faults, for which established defenses, such as secure aggregation [4], or using a Fully Homomorphic Encryption (FHE) and third-party Trusted Execution Environments (TEEs) as in EvoFed [5], remain fully compatible. Notably, MAPO’s low-dimensional update vector substantially reduces both communication and computation costs associated with secure aggregation.
>
> In summary, MAPO seamlessly integrates with existing robustness strategies in FL without requiring modification. We made this clarification in Section 3.
>
> ---
>
> > **Q3:** The labels of Figures 5, 6, and 8 in the article cover up some curves, affecting the readers' observation of the experimental results.
>
> **R3:** Thank you for your attention. We adjusted the Figure legends to improve the visibility of the curves as suggested.
>
> ---
> ```
> [1] Yin, D., Chen, Y., Kannan, R., & Bartlett, P. (2018, July). Byzantine-robust distributed learning: Towards optimal statistical rates. In _International conference on machine learning_ (pp. 5650-5659). Pmlr.
> [2] Blanchard, P., El Mhamdi, E. M., Guerraoui, R., & Stainer, J. (2017). Machine learning with adversaries: Byzantine tolerant gradient descent. _Advances in neural information processing systems_, _30_.
> [3] Zhu, B., Wang, L., Pang, Q., Wang, S., Jiao, J., Song, D., & Jordan, M. I. (2023, April). Byzantine-robust federated learning with optimal statistical rates. In _International Conference on Artificial Intelligence and Statistics_ (pp. 3151-3178). PMLR.
> [4] Bonawitz, K., Ivanov, V., Kreuter, B., Marcedone, A., McMahan, H. B., Patel, S., ... & Seth, K. (2017, October). Practical secure aggregation for privacy-preserving machine learning. In _proceedings of the 2017 ACM SIGSAC Conference on Computer and Communications Security_ (pp. 1175-1191).
> [5] Rahimi, M. M., Bhatti, H. I., Park, Y., Kousar, H., & Moon, J. (2023). EvoFed: leveraging evolutionary strategies for communication-efficient federated learning. Advances in Neural Information Processing Systems, 36, 62428-62441.
>  ```

---

> > ### Comment · Reviewer_b6Vz · 2025-08-06
> >
> > Thank you for your additional input to address my concerns. I will maintain my  rating.

---

### Decision · Program_Chairs · 2025-09-17

**Decision:**

Reject

**Comment:**

This paper provides a communication-efficient heuristic method to speed up distributed training in federated learning. The main technique is to factorize the full gradient into a fixed reconstruction matrix and a projection vector that is trained over iterations. The algorithm is verified experimentally with satisfactory performance - showing its utility. The reviewers agree about the net contribution of this paper. There are some remaining concerns regarding potential instability in the subspace matrix A (as noted in the review by PnwW), and implicit iid condition (noted by reviewer cpPM) which led to mixed enthusiasm about the paper.

In addition a major drawback came up during discussion:  the algorithm generates random iid Gaussian matrices in every iteration and shares it across. This should make the algorithm communication-inefficient. The paper proposes is to deal with this via shared random seeds; however matrices conditioned on such seeds are not iid Gaussian - rendering the analysis questionable. There was a significant deliberation on this point, since the algorithm is a concrete contribution and the question is on theoretical justification; the authors should most certainly clarify this point in the subsequent version of the paper.

Given all these points, this paper is not recommended for acceptance.